# FAST AND EXPRESSIVE MULTI-TOKEN PREDICTION WITH PROBABILISTIC CIRCUITS

## ABSTRACT

Multi-token prediction (MTP) is a prominent strategy to significantly speed up generation in large language models (LLMs), including byte-level LLMs, which are tokeniser-free but prohibitively slow. However, existing MTP methods often sacrifice expressiveness by assuming *independence* between future tokens. In this work, we investigate the trade-off between expressiveness and latency in MTP within the framework of probabilistic circuits (PCs). Our framework, named MTPC, allows one to explore different ways to encode the *joint* distributions over future tokens by selecting different circuit architectures, generalising classical models such as (hierarchical) mixture models, hidden Markov models and tensor networks. We show the efficacy of MTPC by retrofitting existing byte-level LLMs, such as EvaByte. Our experiments show that, when combined with speculative decoding, MTPC significantly speeds up generation compared to MTP with independence assumptions, while guaranteeing to retain the performance of the original verifier LLM. We also rigorously study the optimal trade-off between expressiveness and latency when exploring the possible parameterisations of MTPC, such as PC architectures and partial layer sharing between the verifier and draft LLMs.

## 1 INTRODUCTION

Autoregressive (AR) large language models (LLMs) can only perform single-token prediction (STP) as they generate one token at a time, incurring significantly high latency, energy demand, and deployment costs. This affects not only subword models, but even more so the byte-level ones (Yu et al., 2023; Wang et al., 2024, *inter alia*). Among possible alternatives to speed up generation (Ankner et al., 2024; DeepSeek-AI et al., 2024; Nawrot et al., 2023; Pagnoni et al., 2024; Łańcucki et al., 2025), multi-token prediction (MTP) stands out as it promises to predict a window of multiple tokens *all at once*, may they be subwords (Gloeckle et al., 2024; Cai et al., 2024) or bytes (Gloeckle et al., 2024; Zheng et al., 2025). As such, MTP LLMs can achieve a significantly higher throughput than STP ones, as they decrease the number of forward passes required through the LLM.

Nevertheless, modelling the joint distribution over all future tokens in a window is challenging, as it requires balancing *expressiveness*, i.e., representing all the dependencies between tokens, and *efficiency*, i.e., minimising latency. Existing MTP approaches favour the latter by making an unrealistic assumption: namely, considering all future tokens to be independent (Zheng et al., 2025; Cai et al., 2024; Gloeckle et al., 2024). This clearly comes at the expense of expressiveness (Ankner et al., 2024; Wertheimer et al., 2024), as the choice of a token for a position within the window cannot influence the probability of the others.

For example, consider the prompt: "*Name a capital of South Africa*", where *Cape Town* and *Pretoria* are equally likely completions. A byte-level MTP model with independence assumptions over an 8-token window could return *Cretoria* as an argmax, because replacing *P* with *C* cannot change the probability of other tokens. More concerningly, an exponential number of "byte-salad" continuations, such as *Crptoria*, *Crpt ria* and *Crpt roa*, are then also equally likely, despite having almost zero probability under the STP model. Recently, Basharin et al. (2025) introduced dependencies into MTP with a mixture over the future token probabilities. However, a single mixture can only add limited expressiveness. Crucially, understanding how to increase expressiveness while optimally trading off efficiency in a systematic way is still an open question.

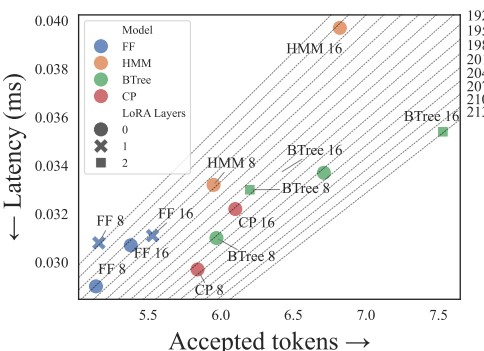

Figure 1: **MTPC allows for exploring the trade-off between efficiency (latency) and expressiveness (token acceptance) with different MTP designs** in terms of 1) choice of PC architecture (FF, CP, HMM, BTree); 2) choice of layers shared between draft and verifier models in self-speculative decoding. Dotted lines indicate iso-throughput (tokens generated per second) regions, highlighting configurations such as BTree for $n = 16$ tokens and 2 LoRA layers that achieve the best throughput.

In this paper, we fill this gap by proposing an MTP framework based on probabilistic circuits (PCs; Choi et al., 2020; Vergari et al., 2021), which we name MTPC. MTPC uses PCs to parameterise the joint distribution over future tokens into tractable computational graphs that can encode hierarchical mixture models. As such, MTPC offers a way to systematically navigate the spectrum of MTP architectural variants, encompassing fully factorised models (Zheng et al., 2025; Cai et al., 2024; Gloeckle et al., 2024) and shallow mixtures (Basharin et al., 2025) but also more expressive parameterisations: hidden Markov models (HMMs) and binary tree factorisations (BTrees) which are novel for MTP.

Moreover, in contrast to previous work on MTP (Zheng et al., 2025; Cai et al., 2024), MTPC guarantees we match the quality of an AR LLM via speculative decoding (Leviathan et al., 2022; Chen et al., 2023; Stern et al., 2018; Xia et al., 2024)—exactly for greedy decoding or in expectation for sampling—showing that the throughput sacrificed for the guarantee is not as large as alluded to previously. We do so by sharing the LLM backbone for the draft and verifier models for different numbers of layers, highlighting how this creates *a second dimension to trade-off expressiveness* (as hidden representations between draft and verifier can diverge) *and latency* (as each non-shared layer requires separate forward passes). We illustrate the two trade-offs at the core of MTPC in Fig. 1.

In summary, we make the following contributions: **C1)** we introduce MTPC, a fast MTP framework based on PCs that overcomes the independence assumptions of previous work and generalises tensor decomposition methods (Basharin et al., 2025); **C2)** we rigorously identify trade-offs between acceptance rates in speculative decoding and latency of generation, based on different choices of probabilistic circuit (PC) architectures and partial layer sharing; **C3)** we empirically demonstrate the effectiveness of MTPC by repurposing EvaByte (Zheng et al., 2025), a byte-level LLM, into our framework. The choice of this use case is motivated by the fact that existing byte-level LLMs (Pagnoni et al., 2024; Wang et al., 2024) obviate the limitations of sub-word tokenisers—including uneven efficiency (Ahia et al., 2023; Dagan et al., 2024), lack of interoperability (Minixhofer et al., 2025), and vulnerabilities (Rumbelow & Watkins, 2023; Land & Bartolo, 2024; Geiping et al., 2024; Salesky et al., 2021)—at the cost of significantly slowing down generation. We find that MTPC increases the throughput of EvaByte by $5.47\times$ with respect to AR generation and $1.22\times$ with respect to MTP with independence assumptions.

## 2 SPEEDING UP GENERATION WITH MTP AND SPECULATIVE DECODING

Given our goal of speeding up LLM generation with MTP while guaranteeing that the STP quality is fully retained through speculative decoding, we introduce these frameworks below.[1]

**MTP.** A classical STP LLM encodes a distribution over sequences of tokens $\{\mathbf{x}_t\}$ defined over a vocabulary $\mathcal{V}$ as $\prod_t p(x_{t+1} \mid \mathbf{x}_{\leq t})$, where $\mathbf{x}_{\leq t}$ is the context, *i.e.* the observed tokens at timestep $t$. MTP (Gloeckle et al., 2024) aims to extend an STP LLM that predicts a single token at a time through $p(x_{t+1} \mid \mathbf{x}_{\leq t})$, to an MTP model, $q_{\boldsymbol{\theta}}$, that models the *joint* probability of a window of $n$ future tokens and generates them *simultaneously*, i.e.,

$$q_{\boldsymbol{\theta}}(x_{t+1}, x_{t+2}, \ldots, x_{t+n} \mid \mathbf{x}_{\leq t}). \tag{1}$$

---

[1]We adapt notation from the tensor and circuit literature (Loconte et al., 2025a), see Appendix A.

where $\boldsymbol{\theta}$ denotes a given parameterisation for the joint.[2] The first dimension to trade-off expressiveness and efficiency in MTP pertains to compactly representing $q_{\boldsymbol{\theta}}$. Unlike for $p(x_{t+1} \mid \mathbf{x}_{\leq t})$, we would need to store more than a vector of logits $\mathbf{a} \in \mathbb{R}^v$ of a single univariate categorical distribution for a vocabulary size $v = |\mathcal{V}|$ for every timestep $t$. The most expressive, but least efficient way to do so, would be to store an $n$-dimensional tensor $\boldsymbol{\mathcal{A}} \in \mathbb{R}^{v^{(1)} \times \dots \times v^{(n)}}$ of logits having $v^n$ entries, but this scales exponentially in $n$. Next, we review past attempts to avoid storing $\boldsymbol{\mathcal{A}}$ explicitly.

**Fully factorised.** The most commonly used way to boost efficiency is to assume all $n$ future tokens are independent (Zheng et al., 2025; Cai et al., 2024; Gloeckle et al., 2024), that is, $q_{\boldsymbol{\theta}}$ factorizes as

$$\prod_{i=1}^{n} q_{\phi_i}(x_{t+i} \mid \mathbf{x}_{\leq t}). \tag{FF}$$

This comes with the benefit that one needs to store only $n$ $v$-dimensional vectors of probabilities $\phi_i$ to represent the joint distribution in Eq. (1). At the same time, as already discussed in the introduction, this severely limits model expressiveness (Ankner et al., 2024; Wertheimer et al., 2024).

**Canonical polyadic (CP) factorisation.** Dependencies between future tokens can be recovered by introducing explicit latent variables (Lee et al., 2018). To this end, Basharin et al. (2025) propose to factorise Eq. (1) via an $r$-rank CP decomposition. A CP decomposition introduces one discrete latent variable, $Z$, that encodes a mixture of $r$ fully-factorised components, rewriting Eq. (1) as:

$$\sum_{j=1}^{r} q(Z = j \mid \mathbf{x}_{\leq t}) \prod_{i=1}^{n} q_{\phi_{i,j}}(x_{t+i} \mid j, \mathbf{x}_{\leq t}). \tag{CP}$$

where $q(j \mid \mathbf{x}_{\leq t}) = \omega_j$ are the mixture coefficients and $\phi_{i,j}$ are the parameters of the categorical distribution for mixture component $j$ at position $i$ in the MTP window. [3] Before showing how we can generalize both FF and CP MTP with PCs, we review how to ensure MTP models match the quality of a given STP model.

**Speculative decoding** (Stern et al., 2018; Leviathan et al., 2022; Chen et al., 2023; Xia et al., 2024) can be combined with MTP to speed up generation while guaranteeing no loss in quality. Given a target STP LLM that we wish to accelerate, speculative decoding involves two steps: 1) *drafting*, where a cheaper MTP draft model generates $n$ future tokens, and 2) *verification*, where the target STP model accepts or rejects the generated tokens in parallel according to a pre-defined consistency criterion. The closer the distributions of the draft and verifier are, the more often 'speculated' tokens are accepted, speeding up generation. With speculative decoding we can quantify the trade-off between expressivenss and efficiency in MTP models as their *throughput*, i.e.

$$\text{throughput (tok/s)} = \text{acceptance rate } \textit{(toks per eval)}/\text{latency } \textit{(secs per eval)} \tag{2}$$

where acceptance rates are a function of the total variation distance between the two distributions (Leviathan et al., 2022; Sun et al., 2023) and latency measures how computationally expensive an MTP model is during generation. While previous work, such as Basharin et al. (2025), focused only on measuring acceptance rates, we highlight how both sides of the ratio in Eq. (2) are important, as they create a spectrum. MTPCs provide a systematic way to navigate such a spectrum (see Fig. 1).

## 3 PROBABILISTIC CIRCUITS FOR MULTI-TOKEN PREDICTION

The idea behind MTPCs is to further decompose the joint distribution in Eq. (1) into a deep computational graph encoding a hierarchical mixture model, called a *probabilistic circuit* (Sections 3.1 and 3.2), and to parameterise it with LLM embeddings (Section 3.3).

### 3.1 PROBABILISTIC CIRCUITS

A ***circuit*** (Darwiche, 2003; Choi et al., 2020; Vergari et al., 2021), $c$, is a parameterised directed acyclic computational graph[4] over variables $\mathbf{X}$ encoding a function, $c(\mathbf{X})$, and comprises three kinds

---

[2]These parameters depend on $t$, we drop the subscript when not needed to avoid clutter.

[3]Basharin et al. (2025) calls CP a mixture of experts (MoE), but we note this is incorrect as the weights $\omega_j$ do not depend on future tokens, but only on past ones. As such, they realise a simple conditional mixture. They argue that training CP is challenging and requires insights from the MoE literature, while we are able to train them as well as deeper mixture variants easily without MoE-tailored losses (see Section 4).

[4]In Fig. 2, edges directionality is removed for readability, but it is assumed to be from inputs to outputs.

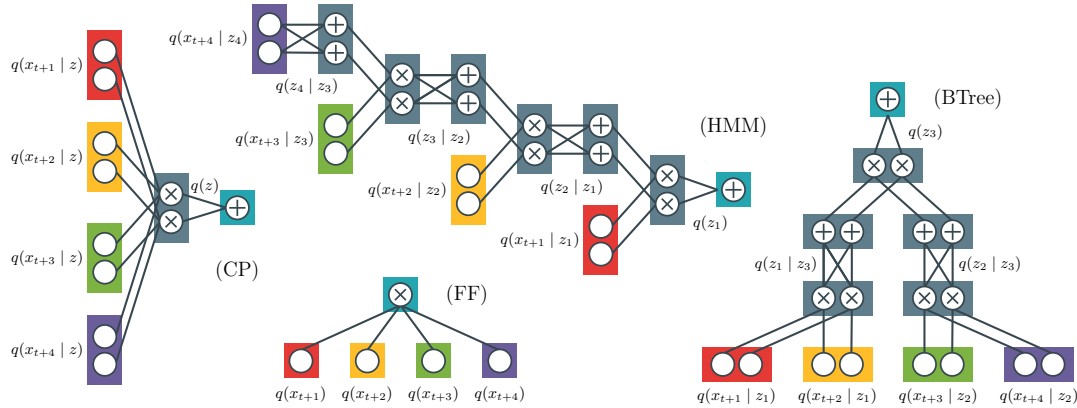

Figure 2: **PCs allow for modelling a spectrum of dependency structures** over sequences of tokens, as shown for the known FF and CP and the novel HMM and BTree MTP variants. Input units are grouped in coloured layers, one for each token, while sum and product layers encoding (hierarchies of) latent variable distributions are in grey. The output unit of each circuit (in blue) computes $q_{\boldsymbol{\theta}}(x_{t+1}, \ldots, x_{t+n} \mid \mathbf{x}_{\leq t})$. In the figure we omit the dependency on the context $\mathbf{x}_{\leq t}$ for readability.

of computational units: *input*, *product*, and *sum* units. Each product or sum unit $n$ receives the outputs of other units as inputs, denoted with the set $\mathsf{in}(n)$. Each unit $n$ encodes a function, $c_n$, defined as: (i) $c_n(\mathsf{sc}(n); \phi)$ if $n$ is an input unit, where $c_n$ is a function parameterised by $\phi$ over variables $\mathsf{sc}(n) \subseteq \mathbf{X}$, called its *scope*; (ii) $\prod_{j \in \mathsf{in}(n)} c_j(\mathsf{sc}(j))$ if $n$ is a product unit; and (iii) $\sum_{j \in \mathsf{in}(n)} \omega_j c_j(\mathsf{sc}(j))$ if $n$ is a sum unit, with $\omega_j \in \mathbb{R}$ denoting the sum parameters. The scope of a product or sum unit $n$ is the union of the scopes of its inputs, i.e., $\mathsf{sc}(n) = \bigcup_{j \in \mathsf{in}(n)} \mathsf{sc}(j)$. Fig. 2 shows examples of circuits, where units of the same scope are grouped into (coloured) layers belonging to a hierarchy that can be easily parallelised on a GPU (Mari et al., 2023; Loconte et al., 2025a).

For MTPCs, we use *probabilistic circuits* (PCs), i.e., circuits modelling a joint distribution over random variables, in our case tokens $\mathbf{X} = \{X_1, \ldots, X_n\}$. PCs encode Eq. (1) as

$$q_{\boldsymbol{\theta}}(x_{t+1}, \ldots, x_{t+n} \mid \mathbf{x}_{\leq t}) = Z_{\boldsymbol{\theta}_t}^{-1} c(x_{t+1}, \ldots, x_{t+n}; \boldsymbol{\theta}_t) \tag{3}$$

where $\boldsymbol{\theta}_t = \{\boldsymbol{\omega}_t, \boldsymbol{\phi}_t\}$ denote the set of circuit parameters, i.e., all sum unit parameters $\boldsymbol{\omega}_t$ and input unit parameterisations $\boldsymbol{\phi}_t$ which depend on the context $\mathbf{x}_{\leq t}$; and $Z_{\boldsymbol{\theta}_t}$ denotes the partition function of $c$, i.e., $Z_{\boldsymbol{\theta}_t} = \sum_{x_{t+1}, \ldots, x_{t+n} \in \mathcal{V}^n} c(x_{t+1}, \ldots, x_{t+n}; \boldsymbol{\theta}_t)$. Note that the PC architectures we are interested in are already normalised or always allow computing the partition function in a single feedforward step (see Choi et al. (2020) and Appendix B.1). At the same time, we can easily sample from PCs in a single feedforward pass, as discussed in Appendix B.2. Crucially, within the framework of PCs, we can recover the FF and CP parameterisations for MTP and several other architectures that generalise tensor factorisations (Loconte et al., 2025a) that can be used as novel MTP models, each offering a different expressiveness-efficiency trade-off. We do so while abstracting away from each model's original formulation and obtain a unified way to parameterise MTP LLMs, as discussed next.

## 3.2 PC Architectures for MTP

**MTPC-FF.** Representing the commonly used FF MTP parameterisation as a PC is simple: we introduce $n$ input units, each parameterised by $\phi_i$, its corresponding token probabilities, and connect them all to a single product unit, as shown in Fig. 2 for a distribution over $n = 4$ tokens.

**MTPC-CP.** Similarly, we can easily encode a CP factorisation in a *shallow* PC by i) introducing $r$ input units for each token (each parameterised by their own probabilities $\phi_{ij}$), then ii) multiplying them to retrieve the $r$ factorised mixture components, which we then iii) aggregate in a sum unit with weights $\omega_j = q(z_j \mid \mathbf{x}_{\leq t})$ (see also Proposition 1 in Loconte et al. (2025a)). Fig. 2 shows this construction for $n = 4$ and $r = 2$. This basic construction suggests that we can create deeper architectures by interleaving sum and product layers, while overparameterising each layer by increasing the number of units in it ($r$). Furthermore, by implementing CP as a PC unlocks a faster sampling routine (Appendix B.2) than the one used in Basharin et al. (2025).

**MTPC-HMM.** As a further example of the expressiveness increase we get by generalising our approach to deeper PCs, we introduce a factorisation that realises a hidden Markov model (HMM), which better captures distant dependencies in the sequence by introducing *a sequence of latent variables*, in contrast to the single one present in CP. More precisely, we define an HMM with $r$ hidden states and truncate its prediction window to $n$ steps into the future. We resort to an inhomogeneous HMM, *i.e.*, we do not make the transition matrices time-invariant, as this setup is more expressive and worked better in our experiments, see Appendix F. This simplifies Eq. (1) into:

$$\sum_{z_1=1}^{r} \cdots \sum_{z_n=1}^{r} q(z_1 \mid \mathbf{x}_{\leq t}) q_\phi(x_{t+1} \mid z_1, \mathbf{x}_{\leq t}) \prod_{i=2}^{n} q(z_i \mid z_{i-1}, \mathbf{x}_{\leq t}) q_\phi(x_{t+i} \mid z_i, \mathbf{x}_{\leq t}). \quad \text{(HMM)}$$

Fig. 2 illustrates the HMM parameterisation above represented as a circuit, comprising $n = 4$ pairs of sum and product layers stacked, where the parameters $\boldsymbol{\omega}_i$ of the former are the transition probabilities $q(z_i \mid z_{i-1}, \mathbf{x}_{\leq t})$. Similarly to CP, we can increase $r$ to overparameterise the circuit with more input units per token and sum units overall, and hence increase expressiveness.

**MTPC-BTREE.** One drawback of the HMM parameterisation is the asymmetry of its computational graph, which i) provides fewer latent variables for the early tokens, and ii) increases latency when predicting the last tokens due to its autoregressive token dependencies. To solve this, we build a PC whose structure resembles that of a binary tree (BTree), effectively encoding a *hierarchy of latent variables* or a tree tensor factorisation (Grasedyck, 2010; Cheng et al., 2019; Loconte et al., 2025a). This is done recursively: at each step $h$ of the hierarchy, given a sequence of $n$ tokens to split, and a parent latent variable $Z_l$, we split it into two sub-sequences $(x_{t+1}, \ldots, x_{t+\lfloor n/2 \rfloor -1})$ and $(x_{t+\lfloor n/2 \rfloor}, \ldots, x_{t+n})$, then factorise Eq. (1) as a mixture:

$$\sum_{z_h=1}^{r} q(z_h \mid z_l, \mathbf{x}_{\leq t}) q_\theta(x_{t+1}, \ldots, x_{t+\lfloor n/2 \rfloor -1} \mid z_h, z_l, \mathbf{x}_{\leq t}) q_\theta(x_{t+\lfloor n/2 \rfloor}, \ldots, x_{t+n} \mid z_h, z_l, \mathbf{x}_{\leq t})$$
$$\text{(BTree)}$$

which corresponds to creating a sum unit whose weights are $q(z_h \mid z_l, \mathbf{x}_{\leq t})$ followed by products. We repeat the process while caching intermediate units until we reach the base case for $n = 1$, for which we create a layer of input units for the corresponding token. Fig. 2 illustrates the BTree circuit built in this way. Our experiments (Section 4.2) show that the BTree parameterisation obtains the optimal throughput by lowering the latency of HMM, as it samples more latent variables and tokens in parallel, while achieving similar acceptance rates.

### 3.3 PARAMETERISING PCs WITH LLMs

Parameterising MTPCs requires two functions: an LLM that maps the context $\mathbf{x}_{\leq t} \in \mathcal{V}^t$ into contextual features, and a neural network head that maps the contextual features to the parameters of the circuit $\boldsymbol{\theta}_t$, realising a *neural conditional circuit* (Shao et al., 2020; 2022; Ahmed et al., 2022). To extract the contextual features $\mathbf{e}_t \in \mathbb{R}^d$, we use $\mathbf{e}_t = \text{LLM}_{\text{LoRA}(k)}(\mathbf{x}_{\leq t})$ where $\text{LLM}_{\text{LoRA}(k)} \colon \mathcal{V}^t \to \mathbb{R}^d$ is the STP backbone with LoRA (Hu et al., 2022) applied to the last $k \geq 0$ layers. As we will discuss in Section 4.4, the number of LoRA layers can impact throughput significantly. Given $\mathbf{e}_t$, we realise Eq. (3) by computing $\boldsymbol{\theta}_t = g_c(\mathbf{e}_t)$, where $g_c$ is a neural network head that outputs both the input unit parameters, $\boldsymbol{\phi}_t$, and the sum unit parameters, $\boldsymbol{\omega}_t$ (Section 3.1). Note that our parameterisation in MTPCs allows us to abstract from the actual structure of the circuit (i.e., FF, CP, HMM or BTree) and just focus on these two sets of tensorised parameters, as we discuss next.

**Input unit distributions.** All MTPCs produce joint distributions over token windows by combining categorical distributions over individual tokens (Fig. 2). We follow EvaByte (Zheng et al., 2025) and learn $n$ separate unembedding layers, one per window position. For models with mixture coefficients, we also learn one unembedding layer per mixture coefficient.[5] As such, instead of a single unembedding matrix mapping $\mathbb{R}^d \to \mathbb{R}^v$, we have an unembedding tensor $\mathcal{W} \in \mathbb{R}^{n \times r \times v \times d}$, and compute the input distributions with the usual unembedding operation followed by softmax, i.e., $\boldsymbol{\phi}_{tij} = \text{softmax}(\mathcal{W}_{ij} \mathbf{e}_t)$, where $i$ and $j$ index the position in the MTP window and the rank $r$.

**Sum unit parameters.** For sum units, instead of mapping embeddings to the vocabulary via $\mathcal{W}$, we map to the rank of the sum unit via $\mathcal{R} \in \mathbb{R}^{z \times r \times d}$, where $z$ is the number of sum units, $r$ is its rank, and $d$ the dimensionality of $\mathbf{e}_t$. We compute $\boldsymbol{\omega}_{ti} = \text{softmax}(\mathcal{R}_i \mathbf{e}_t)$, where $i$ indexes the sum unit.

---

[5]This is efficient even for PCs with high rank due to the small vocabulary size of byte-level LLMs.

### 3.4 SPECULATIVE DECODING WITH MTPC

For MTPCs, we design an architecture that is *self-drafting* (Zhang et al., 2024b; Cai et al., 2024), i.e. where the draft and verifier models share the same LLM backbone. We use an MTP head (Cai et al., 2024; Ankner et al., 2024) augmented with our circuits to efficiently sample a draft, and an autoregressive STP head as the verifier. Optionally, we also explore keeping a few final transformer layers separate in the two models by fine-tuning LoRA adaptors for the draft model's backbone.

Unlike previous self-drafting MTP works (Cai et al., 2024; Ankner et al., 2024), we guarantee that the generated tokens are the same as those the autoregressive LLM would generate in expectation by using *speculative decoding* (Leviathan et al., 2022; Chen et al., 2023), *i.e.*, we only generate the subset of drafted tokens accepted by our verifier. To keep latency low, we make only a single LLM call per speculative decoding cycle by re-using the LLM backbone state computed by the verifier for the draft model, where possible. We achieve this by modifying the speculative decoding algorithm slightly, as we detail in Algorithm 2.

Next we report results for sampling, but we also experimented with greedy speculative decoding (Stern et al., 2018) which guarantees argmax consistency. Both are suitable for MTPCs.

## 4 MTPCS IN ACTION: RETROFITTING A BYTE-LEVEL LLM

We evaluate MTPC on the challenging tasks of speeding up byte-level LLMs. MTP is crucial for byte-level LLMs as they require more tokens than sub-word LLMs to generate text with the same length. Furthermore, byte-level LLMs allow us to explore large window sizes and more mixtures components due to their small vocabulary size. We implement our MTPCs variants in the `cirkit` library (The april Lab, 2024) and provide it in our supplementary materials.

**Target model.** We work with EvaByte (Zheng et al., 2025) as our byte-level LLM, because it is open source, publicly available, and obtains results that are competitive to subword-level LLMs on benchmarks (Zheng et al., 2025), see Appendix C. EvaByte is a 6.5B byte-level model with an embedding size of $4096$, a vocabulary of $320$ byte tokens and a maximum context window of 32k bytes. EvaByte has been pre-trained as an MTP model with a prediction window of $n = 8$ bytes. In our experiments, we retrofit the released fine-tuned version of EvaByte, EvaByte-SFT (Zheng et al., 2025). EvaByte-SFT has been fine-tuned on a data mix of Tülu 3 (Lambert et al., 2024), OpenCoder (Huang et al., 2024) stages one and two, and OpenHermes 2.5.[6] We note that EvaByte's solid performance on benchmarks is obtained via Medusa-style lossy speculative decoding with the MTP head, which in the case of sampling comes with a loss in quality compared to EvaByte-STP ($n = 1$). We therefore set EvaByte-STP as the target model for speculative decoding to *accelerate generation without sacrificing generation quality*.

**Draft models.** We use EvaByte-MTP to refer to EvaByte's released fully-factorised (FF) MTP head. Speculative decoding results have not been reported in the EvaByte release (Zheng et al., 2025), so we include them here as our baseline. We also further fine-tune EvaByte-MTP to highlight that the model cannot be improved further. On top of that, we replace the MTP head with our MTPCs heads, including our CP implementation and novel HMM and BTree heads to relax the independence assumptions of the FF model and increase expressiveness. We note that EvaByte-MTP-CP with $r = 1$ is equivalent to EvaByte-MTP, as can be seen from Eq. (CP).

### 4.1 TRAINING

In order to improve throughput via speculative decoding, we need to make our MTP model's distribution as similar as possible to EvaByte-STP's. We achieve this in the simplest way by instruction fine-tuning our models on a similar data mix to that used for EvaByte-SFT. As the full details of the data mix are not known and are hard to replicate, we focus on Tülu 3.

**Training data.** We fine-tune on the Tülu 3 SFT mix dataset (Lambert et al., 2024) which contains 939,344 examples of user/assistant interactions on 18 tasks. We split the Tülu 3 dataset into training and validation so that we can check throughput on the unseen validation examples. In order to make sure all tasks are sampled, we shuffle the training data before splitting. Because we want training to be

---

[6]The information above is from personal communication with the authors.

possible on $2 \times 80$ Gb GPUs, we limit the context length to 8192 bytes and filter out 34,067 examples which are longer. We split the remaining 905,277 examples into 99% train and 1% validation.

**Initialisation.** We initialise our MTP heads from EvaByte-SFT in a way that guarantees that our EvaByte-MTP-CP is equivalent to EvaByte-MTP. This guarantees that we leverage previous training: all models start from the same loss and we smoothly move in parameter space from EvaByte-MTP to our more expressive EvaByte-MTP-CP, EvaByte-MTP-HMM and EvaByte-MTP-BTree.

**Loss.** We train our MTP models on the packed train split of Tülu 3 with a batch size of 256 sequences, or $\approx 2$m tokens, which is what EvaByte used. We first train our MTP heads for 1 epoch (Section 4.3). Then we load the models and continue training for an additional epoch with LoRA (Section 4.4). We apply EvaByte's chat template and only train on the assistant's answers. We use overlapping prediction windows, as we need to be able to begin speculative decoding from any position during generation. We minimise the negative log-likelihood of the observed assistant outputs Eq. (4), where $N$ is the number of training sequences and $L$ is the sequence length for each token in the window.[7]

$$\mathcal{L} = \sum_{j=1}^{n} \gamma^{j-1} \mathcal{L}_j, \quad \mathcal{L}_j = -\sum_{i=1}^{N} \sum_{t=1}^{L} (\log p_\theta(x_{t+j}^{(i)} \mid \mathbf{x}_{<t+j}^{(i)}))/(N\text{valid}(i,j)) \quad (4)$$

This involves locally normalising the loss by the number of valid tokens for example $i$ and output $j$ in the MTP window, $\text{valid}(i,j)$. As in Cai et al. (2024), we apply exponential discounting for future tokens in the window, but use $\gamma = 0.9$ instead of $\gamma = 0.8$ to account for $n > 8$. We use the Adam optimiser (Kingma & Ba, 2015) with a fixed learning rate of $3 \times 10^{-4}$.

### 4.2 METRICS

To speed up LLMs generation with speculative decoding, we need to balance **speed** and **expressiveness**. We measure speed using **mean latency** ($\mu_{\text{lat}}$) and expressiveness via the **mean acceptance rate** ($\mu_{\text{acc}}$; Li et al., 2024), as defined below. Our goal is to increase **throughput**. We obtain a relative throughput speed-up of one method over another by measuring their **wall-time speedup ratio** (Li et al., 2024; Cai et al., 2024). We assume a batch size of 1 for all evaluations. We report our metrics on two GPUs, the server-grade NVIDIA L40S GPU and the desktop-grade NVIDIA RTX 3090.

**Mean Latency** $\mu_{\text{lat}}$ is the average time taken for each speculative decoding step, *i.e.*, the time needed for the draft model to generate a candidate sequence and the verifier to choose which tokens to accept. $\mu_{\text{lat}}$ is higher for less efficient LLMs and MTP heads, and lower for more powerful GPUs, *e.g.* for EvaByte-MTP the L40S (Tables 1 to 3) has half the latency of the RTX 3090 (Tables 5 to 7).

**Mean acceptance rate** $\mu_{\text{acc}}$ is the percentage of drafted tokens that are accepted by the target model. More expressive draft models will have higher accepance rate as they will better approximate the target distribution. $\mu_{\text{acc}}$ depends on the size of the MTP window, $n$, as we have $\mu_{\text{acc}} \in [0, n]$.

**Mean throughput** $\mu_{\text{tok/s}}$ is measured as in Eq. (2), i.e., as the ratio $\mu_{\text{acc}}/\mu_{\text{lat}}$.

**Wall-time speed-up ratio** is the relative speed-up of a proposed model compared to a baseline model, measured as the ratio of their throughputs. As baselines, we use autoregressive generation from the STP model, EvaByte-STP, and MTP with independence assumptions, EvaByte-MTP FF.

### 4.3 MTPCS WITHOUT ADAPTERS

**RQ1**: Can we increase throughput by increasing the number of mixture components?

We begin with the simplest PC from our framework, MTPC-CP, which relaxes the independence assumption of the widely used MTPC-FF ($r = 1$) by increasing the number of mixture coefficients, $r$. MTPC-CP *can increase throughput because it is more expressive yet still very efficient.*

Table 1 highlights MTPC-CP's efficiency; the $\mu_{\text{lat}}$ introduced by MTPC-CP remains relatively unchanged as we increase $r$, because the forward pass cost of the output layer is dominated by the expensive LLM calls. At the same time, MTPC-CP increases the expressiveness of our MTP head by relaxing the unrealistic independence assumptions. As a result, MTPC-CP with $r = 128$ achieves $\mu_{\text{acc}} = 5.94$, an increase of .82 tokens over MTPC-FF. However, the best throughput is obtained for $r = 32$, where MTPC-CP produces 20.8 more tok/s than MTPC-FF. In the last column, we show

---

[7]Our loss over overlapping windows is a composite log-likelihood (Varin et al., 2011).

| model | $r$ | $\mu_{acc}$ ↑ | $\mu_{lat}$ ↓ | $\mu_{tok/s}$ ↑ | $\max_{tok/s}$ |
|---|---|---|---|---|---|
| FF | 1 | $5.14 \pm 0.06$ | $\mathbf{0.0290} \pm 0.0002$ | $180.1 \pm 2.8$ | $297.50$ |
| CP | 8 | $5.65 \pm 0.02$ | $0.0296 \pm 0.0001$ | $194.5 \pm 1.6$ | $291.61$ |
| | 16 | $5.76 \pm 0.03$ | $0.0299 \pm 0.0002$ | $196.1 \pm 1.9$ | $295.94$ |
| | 32 | $5.84 \pm 0.01$ | $0.0297 \pm 0.0002$ | $\mathbf{200.9} \pm 1.6$ | $292.33$ |
| | 64 | $5.87 \pm 0.09$ | $0.0304 \pm 0.0001$ | $197.2 \pm 2.3$ | $278.42$ |
| | 128 | $\mathbf{5.94} \pm 0.04$ | $0.0320 \pm 0.0001$ | $188.6 \pm 1.1$ | $265.51$ |

Table 1: **Increasing the mixture components ($r$) increases the throughput** ($\mu_{tok/s}$) as seen for MTPC-CP ($n = 8$) over our baseline, EvaByte-MTP (FF) (in gray) where we report the mean $\pm$ std over three sets of 250 prompts. MTPC-CP increases throughput: it has a larger acceptance rate ($\mu_{acc}$) while latency ($\mu_{lat}$) is almost constant in $r$.

| $n$ | $r$ | model | $\mu_{acc}$ ↑ | $\mu_{lat}$ ↓ | $\mu_{tok/s}$ ↑ | speed-up ↑ |
|---|---|---|---|---|---|---|
| 1 | 1 | STP | — | $\mathbf{0.0251}$ | $40.03$ | $1.00$ |
| 8 | 1 | FF | $5.14 \pm 0.06$ | $\mathbf{0.0290} \pm 0.0002$ | $180.1 \pm 2.8$ | $4.50$ |
| | 32 | HMM | $5.95 \pm 0.05$ | $0.0332 \pm 0.0001$ | $182.4 \pm 0.9$ | $4.56$ |
| | 32 | BTree | $\mathbf{5.97} \pm 0.06$ | $0.0310 \pm 0.0004$ | $196.6 \pm 3.8$ | $4.91$ |
| | 32 | CP | $5.84 \pm 0.01$ | $0.0297 \pm 0.0002$ | $\mathbf{200.9} \pm 1.6$ | $\mathbf{5.02}$ |
| 16 | 1 | FF | $5.38 \pm 0.08$ | $\mathbf{0.0307} \pm 0.0004$ | $179.6 \pm 3.8$ | $4.49$ |
| | 32 | HMM | $\mathbf{6.82} \pm 0.04$ | $0.0397 \pm 0.0001$ | $174.5 \pm 0.7$ | $4.36$ |
| | 32 | CP | $6.10 \pm 0.05$ | $0.0322 \pm 0.0001$ | $193.4 \pm 1.8$ | $4.83$ |
| | 32 | BTree | $6.71 \pm 0.01$ | $0.0337 \pm 0.0000$ | $\mathbf{203.5} \pm 0.1$ | $\mathbf{5.08}$ |

Table 2: **More expressive architectures such as MTPC-BTREE outperform MTPC-CP** in terms of throughput for $n = 8$ and $n = 16$ windows on an L40S GPU. For this experiment, we trained only model heads (no LoRAs layers). The shaded baselines are EvaByte-STP and the EvaByte-MTP (FF) models, trained for the same number of steps as our circuits for a fair comparison.

the maximum attainable throughput ($\max_{tok/s}$), *i.e.*, we disable speculative decoding and accept all tokens. The price paid in throughput for guaranteeing no loss in generation quality is $\approx 90$ tok/s for $r = 32$. While MTPC-CP performs well for $n = 8$, the margin for further improving throughput is small. This is because for $n = 8$, we can at best achieve $\mu_{acc} = 8$, and we have already achieved $\mu_{acc} = 5.94$ and have hit diminishing returns. To obtain substantial boosts in throughput, we need to extend our model to longer window sizes. Since $r = 32$ worked best, we keep this fixed for the remaining experiments.

**RQ2:** Do we benefit from more expressive circuit architectures for longer sequences?

We now consider more expressive circuits, such as MTPC-HMM and MTPC-BTREE, and show that they outperform MTPC-CP for longer MTP windows, highlighting the importance of our extension to general PCs. We fix $r = 32$ and explore the different PC architectures for both $n = 8$ and the longer window, $n = 16$. Table 2 shows that MTPC-HMM obtains the best $\mu_{acc}$ in both cases, however, it strikes an unfavourable balance in the expressiveness–latency trade-off: *Due to being AR,* MTPC-HMM *has the largest $\mu_{lat}$, and yields poor throughput as a result.* On the other hand, MTPC-BTREE almost matches the $\mu_{acc}$ of MTPC-HMM and has a smaller $\mu_{lat}$ footprint. Nevertheless, for $n = 8$, MTPC-CP still obtains the best $\mu_{tok/s}$. However, when we move to $n = 16$, MTPC-BTREE substantially increases the gap in $\mu_{acc}$ from MTPC-CP. This in turn leads to MTPC-BTREE having the best throughput, with 203.5 tok/s, a speed-up of $\times 5.08$ over EvaByte-STP. While the gains already obtained by MTPC-BTREE are solid, fine-tuning the output layer alone can only get us so far. This is because EvaByte has not been trained to produce representations that are good for predicting 16 tokens ahead, as we discuss next.

> TAKEAWAY 1: While increasing the mixture components $r$ in CP is initially beneficial, it soon hits diminishing returns. Increasing the window size of future tokens $n$ and adopting more expressive PC architectures unlocks further gains in throughput. Furthermore, while HMM achieves the highest acceptance rates, it incurs high latency. Instead, *non-autoregressive* variants such as BTREE strike a better balance and hence should be preferred.

## 4.4 MTPCs WITH ADAPTERS

**RQ3:** Can we further increase throughput by adapting the draft LLM using LoRA?

| $n$ | model | # LoRA | $\mu_{acc}$ ↑ | $\mu_{lat}$ ↓ | $\mu_{tok/s}$ ↑ | speed-up ↑ |
|---|---|---|---|---|---|---|
| 1 | EvaByte-STP | 0 | — | **0.0251** | 40.03 | 1.00 |
| 8 | MTPC-FF | 0 | 5.15 ±0.04 | 0.0327 ±0.0013 | 163.7 ±10.5 | 4.09 |
|  |  | 1 | 5.16 ±0.02 | **0.0308** ±0.0003 | 171.3 ±1.2 | 4.28 |
|  |  | 2 | 5.14 ±0.06 | 0.0336 ±0.0036 | 157.2 ±17.4 | 3.93 |
|  |  | 4 | 5.19 ±0.03 | 0.0330 ±0.0001 | 160.3 ±1.4 | 4.01 |
|  | MTPC-BTree | 0 | 6.04 ±0.02 | 0.0326 ±0.0027 | 190.5 ±14.8 | 4.76 |
|  |  | 1 | 6.15 ±0.02 | 0.0344 ±0.0038 | 185.1 ±20.3 | 4.62 |
|  |  | 2 | **6.20** ±0.05 | 0.0330 ±0.0000 | **193.0** ±1.4 | **4.82** |
|  |  | 4 | **6.20** ±0.04 | 0.0348 ±0.0001 | 183.1 ±1.0 | 4.57 |
| 16 | MTPC-FF | 0 | 5.40 ±0.06 | **0.0305** ±0.0001 | 180.3 ±2.3 | 4.50 |
|  |  | 1 | 5.53 ±0.08 | 0.0311 ±0.0001 | 182.3 ±2.5 | 4.55 |
|  |  | 2 | 5.63 ±0.07 | 0.0321 ±0.0002 | 179.5 ±2.0 | 4.48 |
|  |  | 4 | 5.60 ±0.03 | 0.0356 ±0.0034 | 162.2 ±14.9 | 4.05 |
|  | MTPC-BTree | 0 | 6.86 ±0.03 | 0.0340 ±0.0001 | 206.1 ±0.9 | 5.15 |
|  |  | 1 | 7.32 ±0.03 | 0.0346 ±0.0000 | 218.0 ±0.6 | 5.45 |
|  |  | 2 | 7.53 ±0.10 | 0.0354 ±0.0001 | **219.1** ±3.0 | **5.47** |
|  |  | 4 | **7.58** ±0.14 | 0.0373 ±0.0003 | 210.2 ±5.0 | 5.25 |

Table 3: **Fine-tuning separate layers in the draft model with LoRA adapters can increase the acceptance rate and speed up BTree MTPCs** for $n = 16$ and two LoRA layers by 5.47 over STP on an L40s GPU. Nevertheless, the increased acceptance rate comes at increased latency, making further throughput boosts via more LoRA layers unviable for EvaByte. We shade the STP baseline in gray and ablated models trained for the additional epoch without LoRA in brown.

We now consider increasing the expressiveness by adding LoRA layers, as shown in Table 3. We show that while we can improve throughput, we need to be strategic when choosing the number of layers, as very quickly the latency introduced outweighs the expressiveness gained.

The key here is that we need to balance the expressiveness obtained by adding LoRA layers and the latency we introduce because the additional layers are not shared between the draft and the verifier. For example, if we train adapters for the last 16 (out of 32) layers, we can improve the acceptance rate by 37%, but we introduce a latency of $1.5\times$ the cost of a forward pass of the LLM.[8] The FF model for $n = 8$ has plateaued, highlighting its limited expressiveness. We highlight that the improvements of MTPC are consistent across GPUs. While throughput is $\approx \times 2$ times larger for the server-grade GPU, the relative speed-ups are similar, see Appendix E. Interestingly, due to the different balance between the LLM and MTPC latency across GPUs, on the RTX 3090 we hit diminishing returns after adding a single LoRA layer rather than two on the L40s.

> TAKEAWAY 2: Fine-tuning a few layers of the draft model with LoRA increases the acceptance rate but also increases latency. The optimal trade-off is device-specific, but adding LoRAs is always beneficial compared with a fully shared LLM trunk. Retrofitting models to longer MTP windows yields an even larger increase in throughput when paired with LoRAs.

## 5 CONCLUSION

Overall, our results show, for the first time, that throughput in MTP LLMs can be increased by $5.47\times$ w.r.t. AR and $1.22\times$ w.r.t. MTP with independence assumptions, while simultaneously guaranteeing the retention of an AR LLM's quality. We achieved this goal by identifying key trade-offs between acceptance rates and latency within our framework, MTPC. We enhanced the *expressiveness* of MTP by getting rid of the independence assumption (Gloeckle et al., 2024; Zheng et al., 2025), introducing an explicit probabilistic model for inter-token dependencies that facilitates performance guarantees (Ankner et al., 2024; Li et al., 2024; DeepSeek-AI et al., 2024), and generalising mixture-based methods (Basharin et al., 2025) into the PC framework. Moreover, we decreased latency by modulating the number of layers shared between draft and verifier model branches. We showcase the throughput gains of MTPC LLMs *at scale* by retrofitting EvaByte (Zheng et al., 2025), a state-of-the-art 6.5B byte-level LLM into our framework.[9] In future work, our framework can be extended by integrating constraints during generation (Ahmed et al., 2025) or speculative decoding (Nakshatri et al., 2025) via methods such as Gelato (Zhang et al., 2023) and Ctrl-G (Zhang et al., 2024a). Unlike those, we would not need to train an auxiliary HMM in MTPCs and we can integrate constraints

---

[8]We found that training more than 16 layers of EvaByte does not lead to improvements in acceptance rates.

[9]More in-depth commentary on related work is available in Appendix I.

directly into our PC head. Moreover, we can further boost expressiveness by leveraging other PCs architectures such as subtractive mixtures (Loconte et al., 2024; 2025b) and continuous latent variable circuits (Gala et al., 2024a;b), while reducing latency through recent advancements in scaling up PCs (Liu et al., 2024; Zhang et al., 2025).

## REPRODUCIBILITY STATEMENT

To ensure reproducibility for our research, we have attached the codebase for implementing all model variants and running their training and evaluation to our submission. In addition, we have provided full details on sampling in circuits in Appendix B and on our algorithms for speculative decoding in Appendix D.

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

## A    NOTATION

We adapt notation and nomenclature from the tensor factorisation (Kolda & Bader, 2009) and circuit (Loconte et al., 2025a) literature.

We denote ordered sets of random variables with $\mathbf{X}$, $\mathbf{Y}$ and $\mathbf{Z}$, and we use $[n]$ to express the set $\{1, 2, \ldots, n\}$ with $n > 0$. The domain of a variable $X$ is denoted as $\mathsf{dom}(X)$, and we denoted as $\mathsf{dom}(\mathbf{X}) = \mathsf{dom}(X_1) \times \cdots \times \mathsf{dom}(X_n)$ the joint domain of variables $\mathbf{X} = \{X_i\}_{i=1}^n$. We denote scalars with lower-case letters (e.g., $a \in \mathbb{R}$), vectors with boldface lower-case letters (e.g., $\mathbf{a} \in \mathbb{R}^N$), matrices with boldface upper-case letters (excluding those used for variables, e.g., $\mathbf{A} \in \mathbb{R}^{M \times N}$), and tensors with boldface calligraphic letters (e.g., $\boldsymbol{\mathcal{A}} \in \mathbb{R}^{I_1 \times I_2 \times I_3}$). Moreover, we use subscripts to denote entries of tensors (e.g., $a_{ijk}$ is the $(i, j, k)$-th entry in $\boldsymbol{\mathcal{A}}$).

## B    BACKGROUND ON CIRCUITS

Circuits have a long history in theoretical computer science (Shpilka & Yehudayoff, 2010) and probabilistic reasoning (Darwiche, 2003; 2009). In their more modern definition and application to machine learning (Vergari et al., 2019b; Choi et al., 2020), circuits are introduced as structured computational graphs, simplified neural networks where one is allowed to use units from a restricted set of neurons (sum, product and input units) and whose connections need to abide certain *structural properties* to guarantee tractability (Choi et al., 2020; Vergari et al., 2021), as discussed next.

### B.1    STRUCTURAL PROPERTIES

Tractability is to be intended as the ability to exactly compute a given function (operation) over the circuit in time that is polynomial in its size, denoted as $|c|$ for a circuit $c$, and representing the number of edges between the computational units. For example, a circuit $c$ can exactly integrate *any subset of variables* in time $\mathcal{O}(|c|)$ if (i) its input functions can be integrated efficiently and (ii) it is *smooth* and *decomposable* (Darwiche & Marquis, 2002; Choi et al., 2020).

**Definition 1** (Smoothness and decomposability (Darwiche & Marquis, 2002; Choi et al., 2020)). A circuit is *smooth* if for every sum unit $n$, all its input units depend on the same variables, i.e., $\forall i, j \in \mathsf{in}(n)\colon \mathsf{sc}(i) = \mathsf{sc}(j)$. A circuit is *decomposable* if the distinct inputs of every product unit $n$ depend on disjoint sets of variables, i.e., $\forall i, j \in \mathsf{in}(n)\ i \neq j\colon \mathsf{sc}(i) \cap \mathsf{sc}(j) = \varnothing$.

Note that all the PC architectures we have discussed in this paper, FF, CP, HMM and BTree, are smooth and decomposable circuits. The reader is encouraged to check this by themselves for the architectures in Fig. 2.

Exactly integrating variables out is relevant to compute marginals such as the normalisation constant of the distribution encoded by the circuit (Eq. (3)). Note that in our implementation, circuits are normalised by design (Peharz et al., 2015), as we assume that input distributions are normalised categoricals and all sum units form a convex combination as their weights are parameterised with a softmax function (see Section 3.3).

More importantly for our MTPCs, we can draw samples efficiently from the distribution of a circuit that is both smooth and decomposable, as we discuss in the next sub-section.

### B.2    SAMPLING A CIRCUIT

A smooth and decomposable PC can use ancestral sampling to generate a complete sample for all $n$ tokens in a window. In a nutshell, we can iteratively sample each latent variable in the hierarchy encoded by the PC, and then sample the selected input distributions, in the same way one sample one (hierarchical) mixture model by first sampling one component and then drawing a sample from that component.

Operationally, Algorithm 1 details the procedure. We have to sample one input branch for each sum unit we encounter when performing a backward traversal of the circuit computational graph (from the circuit output back to the input distributions). Such a branch is sampled proportionally to the sum unit weights $\omega_j$, which encode the mixture components (or equivalently the transition probabilities in

---

**Algorithm 1** SAMPLE($c$)

---

**Input:** A smooth, decomposable and normalised PC $c$ encoding a joint distribution $q$ over the next $n$ tokens $\mathbf{X} = \{X_1, \ldots, X_n\}$ **Output:** a sample $\mathbf{x} \sim q(\mathbf{X})$.

1: $\mathbf{x} \leftarrow$ zeroes($n$)           ▷ init empty sample
2: $c_n \leftarrow$ output($c$)
3: $\mathcal{N} \leftarrow$ queue($\{c_n\}$)       ▷ traverse the computational graph from outputs to inputs
4: **while** $\mathcal{N}$ not empty **do**
5:     $c_n \leftarrow$ pop($\mathcal{N}$)
6:     **if** $c_n = \sum_{j=1}^{r} \omega_j c_j$ **then**       ▷ $c_n$ is a sum unit
7:         $k \leftarrow$ sampleCategorical($\omega_1, \ldots, \omega_r$)    ▷ sample from a categorical with $r$ states
8:         $\mathcal{N} \leftarrow$ push($\mathcal{N}, c_k$)
9:     **else if** $c_n = \prod_{j=1}^{d} c_j$ **then**       ▷ $c_n$ is a product unit with $d$ inputs
10:         **for** $k = 1 \ldots d$ **do**
11:             $\mathcal{N} \leftarrow$ push($\mathcal{N}, c_k$)       ▷ visit all inputs of $c_n$
12:     **else if** $c_n$ is an input unit over variable $X_i$ and parameters $\phi_i$ **then**
13:         $x_i \leftarrow$ sampleCategorical($\phi_i$)    ▷ sample from a categorical with $v = |\mathcal{V}|$ states
14: **return** $\mathbf{x}$

---

an HMM). Then, when we traverse a product unit, we follow all its input branches. When we reach an input unit, we sample a token proportionally to the parameters $\phi_{ij}$ of the categorical distributions encoded in the unit.

If the circuit is smooth and decomposable, by this process we are guaranteed to end up in a set of input units whose scope is the full set of tokens $\mathbf{X}$ and in which only one input unit is selected per token position $i$ (line 13 of Algorithm 1). This procedure can be tensorized as to efficiently generate a batch of samples in a single pass over the computational graph of the circuit (Vergari et al., 2019a; Peharz et al., 2020b;a; Loconte et al., 2025a; Liu et al., 2024).

Lastly, we remark that this routine is potentially computationally more efficient than the one implemented in Basharin et al. (2025), as the latter is based on autoregressive inverse transform sampling (see Loconte et al. (2024) for a discussion) and requires sampling one token at a time.

## C  EVABYTE DETAILS

| Model | BBH | GSM8k | IFEval | MATH | MMLU | HumanEval* | TruthQA |
|---|---|---|---|---|---|---|---|
| Gemma-2-9B-it | 20.0 | 79.7 | 69.9 | 29.8 | 69.1 | 71.7 | 61.4 |
| Ministral-8B-Instruct | 56.2 | 80.0 | 56.4 | 40.0 | 68.5 | 91.0 | 55.5 |
| Qwen-2.5-7B-Instruct | 25.3 | 83.8 | 74.7 | 69.9 | 76.6 | 93.1 | 63.1 |
| Llama-3.1-8B-Instruct | 69.7 | 83.4 | 80.6 | 42.5 | 71.3 | 86.3 | 55.1 |
| Tülu 3 8B | 66.0 | 87.6 | 82.4 | 43.7 | 68.2 | 83.9 | 55.0 |
| OLMo-7B-Instruct | 35.3 | 14.3 | 32.2 | 2.1 | 46.3 | 28.7[†] | 44.5 |
| OLMo-v1.7-7B-Instruct | 34.4 | 23.2 | 39.2 | 5.2 | 48.9 | 49.7[†] | 55.2 |
| OLMoE-1B-7B-0924-Instruct | 37.2 | 47.2 | 46.2 | 8.4 | 51.6 | 54.8 | 49.1 |
| MAP-Neo-7B-Instruct | 26.4 | 69.4 | 35.9 | 31.5 | 56.5 | 72.1[†] | 51.6 |
| OLMo-2-7B-SFT | 50.7 | 71.2 | 68.0 | 25.1 | 62.0 | 67.0[†] | 47.8 |
| OLMo-2-7B-1124-Instruct | 48.5 | 85.2 | 75.6 | 31.3 | 63.9 | 67.6[†] | 56.3 |
| EvaByte-SFT | 34.6 | 52.9 | 60.2 | 29.8 | 49.5 | 73.7 | 46.3 |

Table 4: Downstream benchmark performance of EvaByte-SFT, table taken verbatim from Zheng et al. (2025). Entries with † were computed by the EvaByte authors. All numbers were computed with Medusa-style tree-based greedy decoding using the multi-token head, apart from HumanEval*, which used typical sampling. The authors of EvaByte followed Tulu 3, and evaluated the Pass@10 rate for HumanEval with 20 samples at temperature 0.8.

In Table 4 we provide a copy of the benchmark results of fine-tuned version of EvaByte, EvaByte-SFT, which we retrofit in the paper. The model card can be found at https://huggingface.co/

`EvaByte/EvaByte-SFT`, see also Zheng et al. (2025) for more details. The numbers in the table above were produced by Medusa-style Cai et al. (2024) tree-based typical decoding. For all benchmarks apart from HumanEval, the results were produced via greedy decoding, *i.e.*, similar to (Stern et al., 2018) with the exception of producing the last "free" token. As such, the produced tokens are equivalent to what EvaByte-STP would produce, and the authors found that the metrics were the same with EvaByte-STP up to some small rounding errors. For HumanEval the authors used tree-based typical decoding, which in this case does not maintain the quality of the EvaByte-STP model. The details above were shared with us by Lin Zheng, the author of EvaByte.

## D  SPECULATIVE DECODING

We give pseudocode for our self-speculative decoding algorithm below. The algorithm accepts between $0$ and $n$ tokens, but always generates between $1$ and $n$ tokens, where $n$ is the MTP window size. The algorithm is very similar to vanilla speculative decoding (Leviathan et al., 2022), but our algorithm includes a modification that reduces latency for the self-speculative scenario, and for this it needs to sacrifice the last "free" token typically obtained from the verifier. The gain in latency is possible because we can evaluate the shared LLM once per draft/verify cycle, while a naive implementation of Leviathan et al. (2022) for self-speculative decoding would need two, approximately halving the possible throughput.

In our self-speculative setup, the verifier and draft LLMs share some layers of the backbone. Importantly, the verifier is always computing LLM states ahead of the draft. As such, we can get away with a single forward pass through the shared LLM, similar to Medusa (Cai et al., 2024), by re-using the LLM backbone state computed by the verifier for the draft model. For this to work, we cannot accept a "last sample for free" from the verifier (lines 23-30 Algorithm 3), as we would not have the backbone state for this new token and it is not worth paying an extra LLM evaluation for it. Therefore, in our algorithm we only sample the "free" token from the verifier in the rare case that no tokens are accepted. This is necessary because the model can get caught in successive no-accept states in the sampling case, or get stuck in an infinite loop if we used greedy decoding. If any tokens were accepted, we use the last state of the shared backbone computed during the verify phase to seed the draft phase. In what follows, if we have no LoRA layers, the algorithm is modified to have a single component: the shared encoder.

---

**Algorithm 2** SHAREDSTATESELFSPECULATIVEDECODING

---

**Architecture Components:**
  Three components: Shared Encoder ($S$), Verifier ($V$), Draft ($D$)
  Each with their own KV-cache
**Given:** A prompt of length $L$
**Initialisation:**
Prefill $V$ to $L - 1$, and $D$ and $S$ to $L$
Set $S$ and $D$ state

**Switch to draft/verify cycle:**
**while** true **do**
    **Draft stage:**
    **if** $S$ state is not set **then**
        Compute $S$ by conditioning on the additional token
    Use $S$ state to compute $D$ state
    Parameterize MTPC with $D$ state
    Draft $n$ tokens

    **Verify stage:**
    Compute $S$ state on $n + 1$ tokens (draft + predecessor)
    Compute $V$ state using $S$ state
    Obtain up to $n + 1$ tokens from speculative decoding
    **if** 0 tokens accepted **then**
        Keep "free" token sampled from last valid logits
        Unset $S$ state (stale)
    **else**
        Accept $n$ tokens (drop "free" token)
        Set $S$ state (hidden state for last accepted token)

---

---

**Algorithm 3** SELFSPECULATIVEDECODING($\mathbf{x}_{\leq t}, f, h, c, g$)

---

**Input:** A prefix $\mathbf{x}_{\leq t}$ of length $t$, an LLM backbone $f \colon \mathcal{V}^* \to \mathbb{R}^d$, an LLM head $h \colon \mathbb{R}^d \to \boldsymbol{\Theta}$ parameterising a PC $c$ encoding a joint distribution $q$ over the next $n$ tokens, and an LLM head $g \colon \mathbb{R}^d \to \Delta^v$ computing the next token probabilities.

**Output:** A sentence $(\mathbf{x}_{\leq t} \,\|\, \mathbf{x}_{t+1:t+s}) \in \mathcal{V}^{t+s}$ where $1 \leq s \leq n+1$. Moreover, we have that $\mathbf{x}_{t+1:t+s} \sim p(x_{t+1}, \ldots, x_{t+s} \mid \mathbf{x}_{\leq t})$ as equivalently encoded by the autoregressive single token prediction model consisting of $f$ and $g$ only (Leviathan et al., 2022).

1:    $\mathbf{e}_t \leftarrow f(\mathbf{x}_{\leq t})$                                            ▷ Compute the last embedding

2:    $\boldsymbol{\theta} \leftarrow h(\mathbf{e}_t)$                                         ▷ Compute the circuit parameters

3:

4:    Let $q(\mathbf{X}_{t+1:t+n} \mid \mathbf{x}_{\leq t}) = \frac{1}{Z_{\boldsymbol{\theta}}} c(\mathbf{X}_{t+1:t+n} \mid \boldsymbol{\theta})$

5:    $\mathbf{x}_{t+1:t+n} \sim q(\mathbf{X}_{t+1:t+n} \mid \mathbf{x}_{\leq t})$        ▷ Sample $n$ tokens from $c$ in time $\mathcal{O}(|c|)$

6:    $\mathbf{x} \leftarrow \mathbf{x}_{\leq t} \,\|\, \mathbf{x}_{t+1:t+n}$         ▷ Concatenate the prefix with the $n$ tokens

7:

8:    Compute in parallel for $1 \leq i \leq n$:          ▷ Compute marginals in time $\mathcal{O}(|c|)$

9:       $q(\mathbf{x}_{t+1:t+i} \mid \mathbf{x}_{\leq t}) = \sum_{x_{t+i+1}, \ldots, x_{t+n} \in \mathcal{V}} q(\mathbf{x}_{t+1:t+n} \mid \mathbf{x}_{\leq t})$

10:

11:    Compute in parallel for $1 \leq i \leq n+1$:      ▷ Compute target model conditionals

12:       $p(X_{t+i} \mid \mathbf{x}_{\leq t+i-1}) = g(\mathbf{e}_{t+i-1})$, where $\mathbf{e}_{t+i-1} = f(\mathbf{x}_{\leq t+i-1})$

13:

14:    $s \leftarrow 0$                         ▷ Determine the number of accepted tokens $s$, $0 \leq s \leq n$

15: **while** $s < n$ **do**

16:       $\alpha \sim \mathcal{U}(0,1)$

17:       **if** $s > 0$ **then**

18:           $q(x_{t+s+1} \mid \mathbf{x}_{\leq t+s}) \leftarrow q(\mathbf{x}_{t+1:t+s+1} \mid \mathbf{x}_{\leq t}) / q(\mathbf{x}_{t+1:t+s} \mid \mathbf{x}_{\leq t})$

19:       **if** $\alpha > p(x_{t+s+1} \mid \mathbf{x}_{t+s}) / q(x_{t+s+1} \mid \mathbf{x}_{t+s})$ **then**

20:          **exit loop**

21:       $s \leftarrow s+1$

22:

23:                         ▷ Sample one last token from the autoregressive LLM model

24: **if** $s < n$ **then**              ▷ Adjust the distribution first, if we accept fewer tokens

25:       Let $s(X_{t+s+1}) = q(X_{t+s+1} \mid \mathbf{x}_{\leq t+s})$

26:       Let $m(X_{t+s+1}) = \max\left(0, p(X_{t+s+1} \mid \mathbf{x}_{\leq t+s}) - s(X_{t+s+1})\right)$

27:       $r(X_{t+s+1} \mid \mathbf{x}_{t+s}) = m(X_{t+s+1}) / Z$, with $Z = \sum_{x' \in \mathcal{V}} m(x')$

28:       $x_{t+s+1} \sim r(X_{t+s+1} \mid \mathbf{x}_{\leq t+s})$

29: **else**

30:       $x_{t+s+1} \sim p(X_{t+s+1} \mid \mathbf{x}_{\leq t+s})$

31: **return** $\mathbf{x}_{\leq t+s} \,\|\, x_{t+s+1}$

---

# E  ADDITIONAL RESULTS ON AN RTX 3090

| circuit | $r$ | $\mu_{\text{acc}} \uparrow$ | $\mu_{\text{lat}} \downarrow$ | $\mu_{\text{tok/s}} \uparrow$ | $\max_{\text{tok/s}}$ |
|---------|-----|---------|---------|---------|---------|
| FF | 1 | $5.12 \pm 0.03$ | $\mathbf{0.0519} \pm 0.0003$ | $101.2 \pm 0.1$ | 167.69 |
| CP | 8 | $5.65 \pm 0.03$ | $0.0545 \pm 0.0003$ | $106.3 \pm 0.4$ | 161.49 |
| | 16 | $5.78 \pm 0.06$ | $0.0530 \pm 0.0002$ | $112.2 \pm 1.0$ | 159.22 |
| | 32 | $5.84 \pm 0.04$ | $0.0532 \pm 0.0004$ | $113.1 \pm 0.5$ | 158.55 |
| | 64 | $5.88 \pm 0.03$ | $0.0532 \pm 0.0001$ | $113.8 \pm 0.8$ | 155.04 |
| | 128 | $\mathbf{5.94} \pm 0.03$ | $0.0533 \pm 0.0001$ | $\mathbf{114.8} \pm 0.4$ | 153.24 |

Table 5: **Increasing the mixture components ($r$) increases the throughput** ($\mu_{\text{tok/s}}$) as seen for MTPC-CP ($n = 8$) over our baseline **on an RTX 3090**, EvaByte-MTP (FF) (in gray) where we report the mean $\pm$ std over three sets of 250 prompts. MTPC-CP increases throughput as it has a larger acceptance rate ($\mu_{\text{acc}}$) and a latency ($\mu_{\text{lat}}$) that is constant in $r$.

| $n$ | $r$ | model | $\mu_{\text{acc}} \uparrow$ | $\mu_{\text{lat}} \downarrow$ | $\mu_{\text{tok/s}} \uparrow$ | speed-up $\uparrow$ |
|-----|-----|-------|---------|---------|---------|---------|
| 1 | 1 | STP | — | **0.047** | **21.4** | 1.00 |
| 8 | 1 | FF | $5.12 \pm 0.03$ | $\mathbf{0.0519} \pm 0.0003$ | $101.2 \pm 0.1$ | $\times 4.73$ |
| | 32 | HMM | $\mathbf{5.97} \pm 0.05$ | $0.0594 \pm 0.0001$ | $103.3 \pm 1.0$ | $\times 4.83$ |
| | 32 | BTREE | $5.94 \pm 0.02$ | $0.0546 \pm 0.0005$ | $111.9 \pm 1.1$ | $\times 5.23$ |
| | 32 | CP | $5.84 \pm 0.04$ | $0.0532 \pm 0.0004$ | $\mathbf{113.1} \pm 0.5$ | $\times \mathbf{5.29}$ |
| 16 | 1 | FF | $5.38 \pm 0.03$ | $\mathbf{0.0530} \pm 0.0003$ | $104.3 \pm 1.2$ | $\times 4.87$ |
| | 32 | HMM | $\mathbf{6.81} \pm 0.07$ | $0.0701 \pm 0.0002$ | $99.7 \pm 0.8$ | $\times 4.66$ |
| | 32 | CP | $6.13 \pm 0.03$ | $0.0547 \pm 0.0003$ | $115.4 \pm 1.3$ | $\times 5.39$ |
| | 32 | BTREE | $6.67 \pm 0.07$ | $0.0578 \pm 0.0005$ | $\mathbf{118.9} \pm 2.5$ | $\times \mathbf{5.56}$ |

Table 6: **More expressive architectures such as MTPC-BTREE outperform MTPC-CP** on longer windows in terms of throughput for $n = 8$ and $n = 16$ for no LoRA models **on an RTX 3090 GPU**. We shade the baselines in gray, these are EvaByte-STP and the fully factorised (FF) models trained for the same steps as our circuits.

| $n$ | model | # LoRA | $\mu_{\text{acc}} \uparrow$ | $\mu_{\text{lat}} \downarrow$ | $\mu_{\text{tok/s}} \uparrow$ | speed-up $\uparrow$ |
|-----|-------|--------|---------|---------|---------|---------|
| 1 | STP | 0 | — | **0.047** | **21.40** | **1.00** |
| 8 | FF | 0 | 5.11 | **0.0538** | 97.2 | 4.54 |
| 8 | FF | 1 | 5.09 | 0.0564 | 92.6 | 4.33 |
| 8 | FF | 2 | 5.11 | 0.0567 | 92.6 | 4.33 |
| 8 | FF | 4 | 5.11 | 0.0601 | 87.1 | 4.07 |
| 8 | BTREE | 0 | 6.08 | 0.0568 | **110.0** | **5.14** |
| 8 | BTREE | 1 | 6.15 | 0.0581 | 109.4 | 5.11 |
| 8 | BTREE | 2 | 6.17 | 0.0604 | 105.7 | 4.94 |
| 8 | BTREE | 4 | **6.18** | 0.0625 | 102.3 | 4.78 |
| 16 | FF | 0 | 5.48 | **0.0546** | 102.8 | 4.81 |
| 16 | FF | 1 | 5.55 | 0.0559 | 102.1 | 4.77 |
| 16 | FF | 2 | 5.51 | 0.0584 | 97.2 | 4.54 |
| 16 | FF | 4 | 5.63 | 0.0613 | 94.5 | 4.42 |
| 16 | BTREE | 0 | 6.92 | 0.0587 | 121.4 | 5.67 |
| 16 | BTREE | 1 | 7.26 | 0.0617 | **122.0** | **5.70** |
| 16 | BTREE | 2 | 7.30 | 0.0627 | 120.9 | 5.65 |
| 16 | BTREE | 4 | **7.47** | 0.0669 | 116.2 | 5.43 |

Table 7: **Fine-tuning separate layers in the draft model with LoRA can increase the acceptance rate and speed up BTree MTPCs** for $n = 16$ and one LoRA layer by 5.70 over STP on an **RTX 3090 GPU**. Nevertheless, the increased acceptance rate comes at increased latency, making further throughput boosts via more LoRA layers unviable for EvaByte. We shade the STP baseline in gray and ablated models trained for the additional epoch without LoRA in brown. Interestingly, for the L40S in Table 3, we still got improvements with two LoRA layers, which highlights the importance of carrying out such an analysis across devices.

Table 8: Most Successful HMM Configuration

| Parameterisation | | Transition Type | | Initialisation | |
|---|---|---|---|---|---|
| Contextual | Non-Contextual | Homogeneous | Inhomogeneous | Identity Init. | Uniform Init. |
| ✓ | ✗ | ✗ | ✓ | ✓ | ✗ |

## F  HIDDEN MARKOV MODELS SETUP

In our experiments we use **contextual**, **inhomogeneous** hidden Markov models (HMMs) with **identity initialisation** (see Table 8). We chose the above after preliminary experiments where we assessed the following configuration choices for training our HMMs.

**Parameterisation** We can parameterise HMMs to either be contextual, i.e. we can make the transition probabilities depend on the input, or we can make the transition probabilities be independent of the input (non-contextual).

**Transition Type** The transition matrix can be the same at each time step (homogeneous) or it can be different (inhomogeneous). The former would correspond to additional parameter sharing across sum layers in the circuit representation. We note that inhomogeneous HMMs subsume homogeneous HMMs. This is because inhomogeneous HMMs could in theory learn parameters that do not vary from timestep to timestep, thus becoming equivalent to a homogeneous HMM.

**Initialisation** A crucial setup is to initialise the HMM with transition matrices that are identity matrices, which make the HMM equivalent to CP at the beginning of training. We achieve this by adding a bias term to allow the HMM model to be initialised to identity matrices. This setting in combination with extending to larger token windows, i.e. $n = 16$ lead to a scenario where HMMs outperform CP. The other alternative is to initialise the transition matrices uniformly at random (before softmax), but this complicates learning and yields performance that is lower than CP models.

## G  FURTHER EXPERIMENTAL DETAILS

To make the comparison between methods fair, we: a) constrained the models to not produce end-of-sequence symbols during generation, as the latency of retrieving KV cache items from memory increases with sequence length (Nawrot et al., 2024) and b) we filtered the validation set of the models to only include examples with both prompts and responses in English, as acceptance rates may vary dramatically based on the language chosen for the response.

We compute throughput by generating answers to 250 prompts and report the mean and std of 3 runs with different prompts.

## H  ALTERNATIVE LOSSES

In early versions of this work we also experimented using a Kullback-Leibler divergence (KL) loss as recommended by Basharin et al. (2025). However, we found that training with the KL loss doubled the training time while requiring a lot more memory, and the benefits in acceptance rate did not outweigh the additional complexity. For completeness we include the loss below. **KL Loss** $\mathcal{L}$

$$\mathcal{L} = \sum_{j=1}^{n} \mathcal{L}_j \gamma^{j-1}, \quad \mathcal{L}_j = \sum_{i=1}^{N} \sum_{t=1}^{L} \frac{f_{\text{KL}}\left(p_\theta(x_{t+j}^{(i)} \mid \mathbf{x}_{<t+j}^{(i)}) \,\middle\|\, q_{\theta'}(x_{t+j}^{(i)} \mid \mathbf{x}_{<t+j}^{(i)})\right)}{N \text{valid}(i, j)} \tag{5}$$

In the above we condition both the draft model, $q_{\theta'}$, and the target model, $p_\theta$, on the gold data. The above is equivalent to the KL term from the word-level distillation loss in (Kim & Rush, 2016).[10]

---

[10]While performing sequence-level distillation, *i.e.*, conditioning on data sampled from the teacher model may improve distillation (Kim & Rush, 2016), we did not explore this.

# I  FURTHER RELATED WORK

**MTP for byte-level LLMs** Gloeckle et al. (2024) and Zheng et al. (2025) both pretrain byte-level LLMs which predict $n = 8$ future bytes. This window size was found to be optimal for downstream performance in Gloeckle et al. (2024). Both make conditional independence assumptions, but the approaches are architecturally different. Gloeckle et al. (2024) uses a transformer head per token to provide different feature vectors to each head and uses a shared unembedding matrix. On the other hand, Zheng et al. (2025) uses a shared feature vector across heads with different unembedding matrices per token.[11] However, they only focus on greedy self-speculative decoding, while in our work, we also explore speculative sampling.

**Speculative Decoding with MTP drafts** Previous MTP work either ignores speculative sampling and only focuses on greedy self-speculative decoding (Gloeckle et al., 2024), or abandons guarantees altogether (possibly at the expense of quality): specifically, Cai et al. (2024) and Zheng et al. (2025) use a tree decoding mechanism to consider multiple candidates at each speculative decoding step. Since their approach may accept multiple continuations but only generate the longest accepted one, they bias the distribution and break the guarantees. Wang et al. (2024) use a subword-level draft model to speed up their byte-level STP model via speculative decoding. Most prior work has introduced sequential dependencies in the draft model through architecture modifications. Hydra (Ankner et al., 2024) modifies the Medusa heads such that the predicted probabilities are also a function of the input embeddings of predicted draft tokens. Eagle (Li et al., 2024) introduces sequential dependencies by autoregressively predicting future feature representations. While these works relax the independence assumption, they have no explicit probabilistic model for the dependencies introduced.

An exception is Basharin et al. (2025), who study the effect of relaxing the conditional independence assumption by using a CP factorisation. While they obtain some first promising results, showing that increasing the rank can increase the acceptance rate of tokens for speculative decoding, they focus on subword-level models which have very large vocabulary sizes (e.g. $v \geq 100k$). This makes CP very expensive, both in terms of the number of parameters needed, and the GPU memory required. Moreover, they evaluate their models on unrealistic scenarios, i.e. datasets used for pre-training LLMs rather than instruction fine-tuning. Finally, despite the fact that a lot of previous work exists on MTP for subword-level LLMs, they use different models from those widely used for benchmarking speculative decoding methods, despite the existence of a benchmark, Spec-Bench (Xia et al., 2024) and common models (e.g. Vicuna). In our case, since there is a limited amount of work on MTP for byte-level LLMs (Gloeckle et al., 2024; Zheng et al., 2025), we directly compare with the results of the EvaByte model.

There has been increasing interest in multi-token prediction not only for generation speed-up, but also for improved model performance on tasks due to the lookahead offered by MTP. For example, DeepSeek-AI et al. (2024, Table 4) show that MTP for a token window of 2 tokens leads to improvements in benchmark metrics even when MTP is not used at inference time. Furthermore, they report an increase in the throughput of the model by $\times 1.8$ when using speculative decoding.

**Token granularity** In MTP a token can vary in granularity from bytes (Zheng et al., 2025) to subword tokens provided by tokenisers (Basharin et al., 2025).

In addition to the choice of token granularity, there are generally 3 axes related works differ on:

- Training from scratch vs distilling an existing STP model into a MTP model

- Neural network architectures for the token heads (e.g. Linear, MLP, Transformer)

- Probabilistic modelling assumptions (conditional independence vs more expressive models)

## I.1  DIFFERENCES IN SCENARIO

**Training from Scratch** Evabyte (Zheng et al., 2025) train a MTP byte-level model from scratch using $n = 8$.

---

[11] It is worth noting that Gloeckle et al. (2024) also do an ablation in the appendix and find that linear heads were competitive.

**Retrofitting STP to MTP** Some works explore both training from scratch and retrofitting an STP model into an MTP one (Cai et al., 2024; Basharin et al., 2025). In our work, we focus on the second setting.

### I.2 DIFFERENCES IN ARCHITECTURES

**Linear Heads** Basharin et al. (2025) use a linear parameterisation for each token head in their distillation experiments.

**MLP Heads** Cai et al. (2024) use a MLP with a single hidden layer for each output token head. Ankner et al. (2024) extend this to multiple layers of MLPs per output token. Gloeckle et al. (2024) make the heads context-aware by including a transformer head in each token head.

**Autoregressive Head** While the main point of having future token heads is to avoid the expensive autoregression of the target model, current state of the art speculative decoding models rely on "cheap" autoregression. Eagle (Li et al., 2024), which is the best performing on the speculative decoding benchmark, Spec-bench (Xia et al., 2024), fits an autoregressive model to predict future feature vectors of the model (i.e. the inputs to the softmax layer). A similar architecture was also used for the DeepSeekV3 model (DeepSeek-AI et al., 2024).

**MLP Heads** Cai et al. (2024) propose the Medusa model which uses a MLP with a residual connection for each token head. While in theory they could use many MLP layers, they choose to use MLPs with single hidden layer. Ankner et al. (2024) explore using deeper MLPs for each token head.

**Transformer Heads** Gloeckle et al. (2024) use a shared unembedding matrix and use a separate transformer for each token head. More precisely, in order to predict the token at offset $s$, i.e. $x_{t+s}$, they compute $\mathrm{softmax}\left(\mathbf{W}\mathbf{z}_s\right)$, where $\mathbf{z}_s$ is produced by a separate transformer head for each $s$, i.e. $\mathbf{z}_s = H_s(\mathbf{x}_{\leq t})$.

**Sharing the Unembedding layer.** While the decoding of Zheng et al. (2025) is based on Medusa, instead of using a MLP for each token head, they use a different unembedding matrix per token head.[12] This modelling choice is possible due to the small vocabulary size of byte-level models, i.e. $|\mathcal{V}| = 320$. In their training from scratch scenario, Basharin et al. (2025) use different unembedding layers $\mathbf{W}_a^{(s)}$ in order to predict $x_{t+s}$ for the mixture component with index $a$. As such, they parameterise $s \times |a|$ unembedding matrices. This seems non-ideal, since the last layer in LLMs can have a large number of parameters, i.e. $(V \times d)$. In their distillation scenario they use a shared unembedding matrix.

## J  LLAMA MODEL RESULTS

**Llama3.2 3B**   We also retrofit a Llama3.2 3B model (Grattafiori et al., 2024) to show that our results generalise and are useful for popular models and models of smaller size. Since we focus on byte-level LLMs, we retrofit the byte-level version that has been distilled from the original in (Minixhofer et al., 2025) while retaining most of Llama's downstream performance.[13] The byte-fied Llama3.2 3B model was fine-tuned on Tülu 3 with a context length of 2048.

Below we corroborate our EvaByte findings for **RQ1** and **RQ2** by showing that our results for the byte-fied version of Llama 3.2 3B lead to the same conclusions: i) increasing the number of mixture coefficients increases the acceptance rate and therefore the throughput, and ii) more expressive circuits like MTPC-BTREE further improve throughput over MTPC-CP. As with our main EvaByte results, we use $n = 8$ and do not use LoRA layers. We run all speculative decoding experiments on an NVIDIA L40s GPU.

**RQ1**:  Can we increase throughput by increasing the number of mixture components?

As can be seen in Table 9, similarly to our EvaByte results, increasing the rank leads to increased acceptance rates and throughput for Llama3.2.

**RQ2:**  Do we benefit from more expressive circuit architectures (for longer sequences)?

---

[12]https://github.com/OpenEvaByte/evabyte/blob/98d5f48d32197b803e7560a798da35c7a4bdcf4d/evabyte_hf/modeling_evabyte.py#L753

[13]https://huggingface.co/benjamin/Llama3-2-3B-IT-Byte

As can be seen in Table 10, for the Llama model we get the best performance using MTPC-BTREE even for $n = 8$, while for EvaByte, MTPC-CP had superior throughput. Although we get smaller relative speed-ups over STP $(2.07\times)$ when compared to EvaByte, we still get substantial boosts in throughput by relaxing the independence assumptions, with MTPC-CP increasing throughput by $1.14\times$ and MTPC-BTREE by $1.23\times$ over MTPC-FF. We note that EvaByte has been pretrained with an MTP head, so it is reasonable that it is hard to match its throughput while keeping the Llama encoder frozen (no-lora). We also note that for the Llama model the generation quality of the STP model degrades after 500 tokens or so when using ancestral sampling. We believe this is why the greedy decoding results in Tables 11 and 12 are even better compared to the sampling results.

## J.1 SPECULATIVE SAMPLING

| model | $r$ | $\mu_{\text{acc}}\uparrow$ | $\mu_{\text{lat}}\downarrow$ | $\mu_{\text{tok/s}}\uparrow$ |
|---|---|---|---|---|
| FF | 1 | $1.73_{\pm 0.02}$ | $\mathbf{0.0516}_{\pm 0.0006}$ | $36.1_{\pm 0.6}$ |
| CP | 8 | $1.92_{\pm 0.07}$ | $0.0529_{\pm 0.0018}$ | $39.3_{\pm 0.5}$ |
| | 16 | $1.98_{\pm 0.02}$ | $0.0550_{\pm 0.0011}$ | $38.9_{\pm 0.8}$ |
| | 32 | $\mathbf{2.08}_{\pm 0.04}$ | $0.0543_{\pm 0.0012}$ | $\mathbf{41.1}_{\pm 0.7}$ |

Table 9: Speculative sampling results while varying the CP rank for the byte-fied version of Llama 3.2 3B. As can be seen, increasing $r$ steadily increases the acceptance rate and the best throughput is obtained for $r = 32$.

| $n$ | $r$ | model | $\mu_{\text{acc}}\uparrow$ | $\mu_{\text{lat}}\downarrow$ | $\mu_{\text{tok/s}}\uparrow$ | speed-up $\uparrow$ |
|---|---|---|---|---|---|---|
| 1 | 1 | STP | — | $\mathbf{0.049}$ | $21.5$ | $1.00$ |
| 8 | 1 | FF | $1.73_{\pm 0.02}$ | $0.0516_{\pm 0.0006}$ | $36.1_{\pm 0.6}$ | $1.68$ |
| | 32 | CP | $2.08_{\pm 0.04}$ | $0.0543_{\pm 0.0012}$ | $41.1_{\pm 0.7}$ | $1.92$ |
| | 32 | BTree | $\mathbf{2.26}_{\pm 0.02}$ | $0.0549_{\pm 0.0013}$ | $\mathbf{44.3}_{\pm 1.2}$ | $\mathbf{2.07}$ |

Table 10: Speculative sampling results with $r = 32$ for the byte-fied version of Llama 3.2 3B. As can be seen, MTPC-BTREE outperforms all other models in throughput, supporting our EvaByte results.

## J.2 GREEDY SPECULATIVE DECODING

| model | $r$ | $\mu_{\text{acc}}\uparrow$ | $\mu_{\text{lat}}\downarrow$ | $\mu_{\text{tok/s}}\uparrow$ |
|---|---|---|---|---|
| FF | 1 | $2.70_{\pm 0.01}$ | $\mathbf{0.0463}_{\pm 0.0018}$ | $59.8_{\pm 2.5}$ |
| CP | 8 | $3.25_{\pm 0.02}$ | $0.0490_{\pm 0.0022}$ | $69.4_{\pm 3.2}$ |
| | 16 | $3.42_{\pm 0.04}$ | $0.0502_{\pm 0.0018}$ | $71.3_{\pm 2.4}$ |
| | 32 | $\mathbf{3.47}_{\pm 0.06}$ | $0.0507_{\pm 0.0015}$ | $\mathbf{71.9}_{\pm 3.0}$ |

Table 11: Greedy speculative decoding results while varying the CP rank for the byte-fied version of Llama 3.2 3B.

| $n$ | $r$ | model | $\mu_{\text{acc}}\uparrow$ | $\mu_{\text{lat}}\downarrow$ | $\mu_{\text{tok/s}}\uparrow$ | speed-up $\uparrow$ |
|---|---|---|---|---|---|---|
| 1 | 1 | STP | — | $\mathbf{0.049}$ | $21.5$ | $1.00$ |
| 8 | 1 | FF | $2.70_{\pm 0.01}$ | $0.0463_{\pm 0.0018}$ | $59.8_{\pm 2.5}$ | $2.79$ |
| | 32 | CP | $3.47_{\pm 0.06}$ | $0.0507_{\pm 0.0015}$ | $71.9_{\pm 3.0}$ | $3.35$ |
| | 32 | BTree | $\mathbf{3.72}_{\pm 0.09}$ | $0.0507_{\pm 0.0017}$ | $\mathbf{76.8}_{\pm 4.1}$ | $\mathbf{3.58}$ |

Table 12: Greedy speculative decoding results with $r = 32$ for the byte-fied version of Llama 3.2 3B.

