# OpenReview forum: "Fast and Expressive Multi-Token Prediction with Probabilistic Circuits"
_ICLR.cc/2026/Conference — Submitted to ICLR 2026_

### Official Review · Reviewer_VKHG · 2025-10-31

**Soundness:** 3
**Presentation:** 2
**Contribution:** 2
**Rating:** 4
**Confidence:** 4

**Summary:**

This paper introduces MTPC, a probabilistic-circuit-based framework for multi-token prediction (MTP) in large language models. Unlike traditional MTP approaches that assume independence among future tokens, MTPC flexibly encodes joint token distributions through diverse probabilistic circuit architectures, generalizing models such as mixture models, HMMs, and tensor networks. Applied to byte-level LLMs like EvaByte, MTPC, combined with speculative decoding, substantially accelerates generation while preserving the verifier model’s performance. The study systematically explores the trade-off between expressiveness and latency, showing that appropriate architectural and parameter-sharing choices yield efficient, expressive, and consistent multi-token generation.

**Strengths:**

1. Theoretical novelty: The paper provides an interesting and elegant theoretical formulation of multi-token prediction using probabilistic circuits, enriching understanding of the expressiveness–latency trade-off.

2. Significant performance gains: Experimental results demonstrate strong acceleration and efficiency improvements while maintaining model quality, highlighting the practical impact of the proposed approach.

**Weaknesses:**

1. Lack of comparison with related methods: The paper does not clearly distinguish MTPC from tree-based speculative decoding approaches, nor does it discuss relevant prior work, which weakens its positioning in the broader MTP literature. [1]https://arxiv.org/abs/2402.12374 [2] https://arxiv.org/abs/2305.09781 [3] https://arxiv.org/abs/2401.10774

2. Limited model applicability: The experiments focus mainly on byte-level LLMs, with no evaluation or discussion on generalizing MTPC to mainstream models such as LLaMA, Qwen, or DeepSeek, leaving its scalability and universality uncertain.

**Questions:**

In the weakness.

---

> ### Author Response · Authors · 2025-11-21
>
> We thank the Reviewer for their clear feedback and helpful pointers to prior work. We are elated that they found our theoretical formulation elegant and our results strong, and that they are convinced our analysis enriches our understanding of the expressiveness-latency trade-off for speculative decoding.
>
> > [W1] Lack of comparison with related methods: The paper does not clearly distinguish MTPC from tree-based speculative decoding approaches, nor does it discuss relevant prior work, which weakens its positioning in the broader MTP literature. \[1]<https://arxiv.org/abs/2402.12374> \[2] <https://arxiv.org/abs/2305.09781> \[3] <https://arxiv.org/abs/2401.10774>
>
> We agree that tree-based speculative decoding approaches are very interesting and achieve great improvements in throughput. However, **we highlight that tree-based speculative decoding is compatible but also orthogonal to our work**: our goal is not to speed up speculative decoding in any way possible. Instead we highlight the expressivity/latency trade-off by using principled probabilistic models and guaranteeing no drop in text generation quality such that expressivity is as quantifiable as possible.
>
> However we agree that **we can more clearly distinguish MTPC from tree-based speculative decoding approaches** and thank the Reviewer for the pointers to related work, we plan to incorporate these into the paper by the end of the discussion period.
>
> > [W2] Limited model applicability: The experiments focus mainly on byte-level LLMs, with no evaluation or discussion on generalizing MTPC to mainstream models such as LLaMA, Qwen, or DeepSeek, leaving its scalability and universality uncertain.
>
> **We will address this limitation by retrofitting byte-fied models such as Gemma and Llama**, please also see the general response.

---

### Official Review · Reviewer_azfz · 2025-10-31

**Soundness:** 3
**Presentation:** 3
**Contribution:** 3
**Rating:** 6
**Confidence:** 2

**Summary:**

The paper proposes MTPC, a family of multi-token prediction (MTP) heads built from probabilistic circuits (FF, CP, HMM, BTree) that model joint distributions over future tokens and plug into a shared-backbone self-speculative decoding setup. The framework lets one trade off expressiveness (acceptance rate) vs latency by (i) choosing the PC architecture and (ii) selecting how many layers the draft/verifier shares or separates. On EvaByte (byte-level LLM), MTPC improves throughput over AR and over fully factorized MTP while guaranteeing AR quality under speculative decoding.

**Strengths:**

1. The paper introduces MTPC, a multi-token prediction framework built on probabilistic circuits, which overcomes the independence assumptions of prior MTP methods. This allows MTPC to model joint token dependencies more effectively than factorized or tensor-decomposition-based approaches.

2. The paper rigorously studies the trade-offs between acceptance rate and generation latency across different PC architectures and different levels of layer sharing. This provides a clear and interpretable design space for controlling speed–quality trade-offs.

3. The framework is evaluated on EvaByte, where MTPC demonstrates substantial throughput improvements, for example, ×5.47 over autoregressive decoding and ×1.22 over MTP models with independence assumptions, while maintaining output quality. The experiment highlights practical deployment viability in real LLM inference settings.

**Weaknesses:**

1. Experiments focus on a single 6.5B byte-level model (EvaByte) and one SFT mixture (Tülu-3). It would strengthen claims to show transfer to a subword LLM (to decouple gains from byte vocabularies) and to other additional datasets/domains.

2. While the loss and discounting are described, ablations on optimization sensitivity (γ, window overlap, head depth/width) are limited. Providing more ablation studies would strengthen the paper.

3. The paper emphasizes latency but gives fewer numbers on memory vs n and r for different PCs (esp. BTree with higher ranks). Besides, this paper introduces a verifier that consumes an additional memory footprint. Therefore, a detailed memory footprint plot will help the audience understand the memory consumption of this paper.

**Questions:**

See weaknesses.

---

> ### Author Response · Authors · 2025-11-21
>
> We thank the Reviewer for their insightful comments, and for highlighting several strengths of our method, including the ability to jointly model future token dependencies, a thorough analysis of the latency-expressivity tradeoffs, and the substantial gains in throughput.
>
> We address the weaknesses identified by the Reviewer below:
>
> > [W1] Experiments focus on a single 6.5B byte-level model (EvaByte) and one SFT mixture (Tülu-3). It would strengthen claims to show transfer to a subword LLM (to decouple gains from byte vocabularies) and to other additional datasets/domains.
>
> **We plan to run additional experiments that retrofit other LLMs to showcase the generality of our method**. Unfortunately, adapting MTPC to subword-level LLMs as-they-are is beyond the scope of this work, as it would entail different token windows and numbers of mixture components: the vocabulary size (hence, the memory load) would be significantly larger, entailing different latency–efficiency tradeoffs and would be outside our computational budget. However, Minixhofer et al. \[1] recently showed that subword-level LLMs (such as Gemma and Llama family) can be turned into byte-level LLMs with no significant loss of quality. Therefore, we can always start with a subword LLM, turn it into a byte-level one, and then boost generation speed with it further with our MTPC framework. In the remainder of the discussion period, we will retrofit these byte-fied LLMs to show similar throughput gains in SOTA LLMs.
>
> > [W2] While the loss and discounting are described, ablations on optimization sensitivity (γ, window overlap, head depth/width) are limited. Providing more ablation studies would strengthen the paper.
>
> **We ablated some of these settings in early experiments using the setup from Basharin**. We found that γ =.8 was better than using γ=1 (i.e. no exponential discounting). We also ablated head depth: we replaced heads with MLPs like those used in Medusa but observed no difference in performance, so we resorted to Linear heads. In terms of width, the unembedding matrices for EvaByte are square (320 x 320) and therefore as expressive as possible.
>
> We agree that further ablations would strengthen the paper and agree that the suggested ablations open up interesting research questions, such as: what is the optimal window overlap and can we subsample windows to speed up training? However, these experiments are orthogonal to the research questions we answer in the paper and while they may improve our results and convey more experimental information, they are unlikely to change our findings. We are willing to run other ablations for the camera-ready if the reviewer can suggest other specific ones.
>
> > [W3] The paper emphasizes latency but gives fewer numbers on memory vs n and r for different PCs (esp. BTree with higher ranks). Besides, this paper introduces a verifier that consumes an additional memory footprint. Therefore, a detailed memory footprint plot will help the audience understand the memory consumption of this paper.
>
> We thank the reviewer for their thoughtful suggestion that will help readers better understand the memory trade-off as we increase the rank of MTPC models. **We are currently running an analysis of GPU memory usage during inference for BTree and CP as we increase the rank r and n \[8, 16] to further inform our results**. We will include these in the updated paper before the end of the discussion period.
>
> We also note that memory is not a bottleneck for our trained models. For the retrofitted models with LoRA layers (RQ3) we have optimised our speculative code to share activations and model parameters where possible (e.g. we only pay additional memory for the LoRA layer parameters and their activations when compared to the shared encoder case). All the models we trained fit on an NVIDIA 3090 GPU with 24 Gb of RAM when generating using speculative decoding.
>
> \[1] Cross-Tokenizer Distillation via Approximate Likelihood Matching <https://arxiv.org/abs/2503.20083>

---

> > ### Comment · Reviewer_azfz · 2025-11-25
> >
> > Thank you for your response. I have no further questions.

---

### Official Review · Reviewer_jubx · 2025-11-01

**Soundness:** 3
**Presentation:** 4
**Contribution:** 3
**Rating:** 4
**Confidence:** 4

**Summary:**

This paper proposes MTPC, a framework for MTP in LLMs based on probabilistic circuits. Existing MTP methods assume independence between future tokens, sacrificing expressiveness and leading to implausible outputs. MTPC addresses this by parameterizing joint distributions over token windows using PC architectures that encode hierarchical mixture models. The framework encompasses fully factorized models (FF), canonical polyadic decompositions (CP), and introduces novel hidden Markov model (HMM) and binary tree (BTree) factorizations for MTP. Combined with speculative decoding, MTPC guarantees retention of the original autoregressive LLM's quality. The authors identify two key trade-offs: (1) PC architecture choice affecting expressiveness vs. latency, and (2) number of LoRA layers shared between draft and verifier models. Experiments retrofitting EvaByte (a 6.5B byte-level LLM) demonstrate 5.47× speedup over autoregressive generation and 1.22× speedup over independence-based MTP, with BTree achieving optimal throughput for n=16 tokens and 2 LoRA layers.

**Strengths:**

1. MTPC provides a unified probabilistic circuit framework that systematically navigates MTP design space, introducing novel HMM and BTree architectures with BTree achieving optimal throughput by parallelizing latent sampling while maintaining high acceptance rates.
2. The paper rigorously examines trade-offs across PC architecture selection (FF/CP/HMM/BTree) and partial layer sharing via LoRA (0-4 layers), revealing device-specific optimal configurations through systematic ablations across mixture components, window sizes, and GPU types.
3. MTPC uses speculative decoding to provably match autoregressive quality while achieving 5.47x speedups, outperforming provided baselines.

**Weaknesses:**

1. All experiments focus exclusively on EvaByte (6.5B byte-level model with v=320), without validation on subword-level LLMs where vocabularies are 300× larger or across different model families/sizes, limiting claims about scalability.
2. Key design decisions including inhomogeneous HMMs, identity matrix initialization, and why BTree outperforms CP lack theoretical justification beyond empirical validation, with no analysis of when specific architectures excel for different prompt characteristics.
3. The paper omits comparisons with recent MTP methods like Hydra and Eagle that introduce sequential dependencies, dismisses Basharin's KL loss without thorough evaluation, and lacks validation on standard speculative decoding benchmarks.

**Questions:**

1. Have authors evaluated MTPC on subword-level LLMs with large vocabularies (v≥100k), and how do the memory/computational costs of CP/HMM scale compared to FF in such settings?
2. Can authors provide theoretical or empirical guidelines for when to choose BTree vs. HMM vs. CP based on prompt characteristics, sequence lengths, or task requirements beyond throughput measurements?

---

> ### Author Response · Authors · 2025-11-21
>
> We thank the Reviewer for their thoughtful review. We are excited that they found our work novel, appreciated our presentation of a unified framework for MTP via circuits and valued our rigorous examination of trade-offs across PC architectures and partial layer sharing.
>
>
> > [W1] All experiments focus exclusively on EvaByte (6.5B byte-level model with v=320), without validation on subword-level LLMs where vocabularies are 300× larger or across different model families/sizes, limiting claims about scalability.
>
> We agree that our paper would benefit from additional experiments on subword-level models, but GPU memory is a bottleneck (please see our answer to Q1). While we still observed improvements in acceptance rate for subword level models, we found that the amount of engineering needed to optimise them detracted from our research questions which are about the expressivity/latency trade-off. At the same time, **we did not make any claims about scalability w.r.t the vocabulary size in our paper, so we respectfully disagree that any of our claims are invalidated**.
>
> Nevertheless, we agree that our claims would be stronger if we evaluated widely-used subword-level models and included an LLM of a different size. We highlight that one of the reasons byte-level models are rarely used today is their slow decoding. Our work has the potential to change that. **We are therefore running experiments on a Llama 3.2 3B model (half the size) that has been cross-tokeniser distilled to a byte-level model while retaining most of the performance of the original subword-level mode**l, see <https://huggingface.co/benjamin/Llama3-2-3B-IT-Byte>.
>
> > [W2] Key design decisions including inhomogeneous HMMs, identity matrix initialization, and why BTree outperforms CP lack theoretical justification beyond empirical validation, with no analysis of when specific architectures excel for different prompt characteristics.
>
> We thank the Reviewer for pointing out that some of our design decisions were not as clearly articulated and justified as we intended. We note that **many of these decisions were explained in the “Hidden Markov Models Setup” in the appendix** (see lines **1088-1103**), e.g. the HMM initialisation question is answered in the “initialisation” paragraph. We have expanded upon this section in the updated version of the paper and answer in more detail below.
>
> **Inhomogeneous HMMs** are more expressive than homogeneous HMMs, because they subsume them: an inhomogeneous HMM could in theory learn parameters that do not vary from timestep to timestep, thus becoming equivalent to a homogeneous HMM.
>
> **BTree outperforms CP** because it is more expressive. BTree has more latent variables than CP and these latent variables are more localised, i.e. the leaf latent variables of a BTree model a subset of the MTP window, while the CP latent variables model the full window. This can be seen from the generative story which is illustrated in Figure 2 in the paper.
>
> **Identity matrix initialisation** for HMMs is theoretically justified because it is equivalent to initialising a model to CP - which is the simplest model, see **line 1101**.
>
> > [W3] The paper omits comparisons with recent MTP methods like Hydra and Eagle that introduce sequential dependencies, dismisses Basharin's KL loss without thorough evaluation, and lacks validation on standard speculative decoding benchmarks.
>
> **We respectfully disagree that we should compare to Hydra and Eagle since the main purpose of our paper is not to propose a new speculative decoding algorithm**, but rather 1) to highlight how the expressivity/efficiency trade-off is the ideal perspective through which to design faster MTP methods, which is not clear-cut in previous work; 2) to operate in a setting that provides guarantees that the text quality is not diminished wrt the autoregressive model. Moreover, Hydra and Eagle use cheap autoregression and are therefore not Multi-Token-Prediction models in the true sense of the concept: they still produce one token at a time and are therefore not as parallelisable as true MTP models. At the same time, we have not evaluated our models on speculative decoding benchmarks like spec-bench because the comparison to subword-level LLMs is an apples and oranges comparison that we believe would not improve our analysis.
>
> > Q1
>
> In preliminary experiments we retrofitted the Llama model Basharin used in their experiments (v=50272) and set n=2 and r=4. We found that GPU memory was a bottleneck due to the large vocabulary size: it was hard to scale beyond r=4 for our resources (2 NVIDIA-A100-SXM4-80GB GPUs) when using a context length of 256 because the number of logits scale linearly in r.
>
> > Q2
>
> While this question is interesting, it is hard for us to say in general which model will perform the best for metrics that are unrelated to throughput and this investigation would be out of scope for our paper, since we focus on speculative decoding.

---

### Official Review · Reviewer_EgHu · 2025-11-04

**Soundness:** 3
**Presentation:** 3
**Contribution:** 3
**Rating:** 6
**Confidence:** 4

**Summary:**

This paper introduces multi-token PCs (MTPCs). The main idea is to movie beyond fully-factorized and simple mixture models in the context of multi-token prediction for speculative decoding. The authors evaluate MTPC on EvaByte, a byte-level LLM and observe that MTPC increases the throughput of EvaByte by 1.22x compared to the less expressive MTP speculative decoding.

**Strengths:**

- I found the paper to overall be well-written, aside from a few nitpicks that I've highlighted in my questions below.

- The paper offers a general, principled framework that encompasses several of the previous works.

- By exploiting connections to previous work, the authors manage to increase the expressiveness of the drafters while minimizing the latency for an overall improved throughput of 1.22x

**Weaknesses:**

- The paper deals with byte-level LLMs which in my opinion greatly limits its scope as it's hard to draw strong conclusion about its performance on sub-word LLMs that are a lot more commonly used by the community.

- The paper details the requirement to train the MTPC which by the authors' description is a very arduous process, and could therefore
limit adoptability of the proposed approach.

**Questions:**

- The authors mention that "MTPC guarantees that they match the quality of an AR LLM via speculative decoding, exactly for greedy decoding, or in expectation for sampling". Are the authors making the claim that the output of MTPC follows the AR LLM distribution? If so, isn't that a standard assumption in speculative decoding approaches? Is the "in expectation for sampling" a weakening of that assumption?

- The authors mention "repurposing" and/or "retrofitting" EvaByte, but my understanding is that the language modeling component is largely left unchanged?

- I find it a bit confusing how *speculative decoding* is separated from the *fully-factorized* and *canonical polyadic factorization* in section 2, since it is my understanding that the latter two are a means to realizing the former.

- I believe the parameterization of the PC with an LLM bears great resemblance to [1], which should be mentioned.

- I am a bit confused by paragraph 293-303. Is the implication that the model being used, EvaByte, is used with n=1 to recover a STP model? Are all the experimental results reported using greedy decoding with EvaByte? If so, it would've been useful to expand more upon the greedy speculative decoding paper by Stern et. al to show how one can guarantee argmax consistency with speculative decoding (which is not specific to MTPC)

- Referencing your conclusion, similar to the work of Zhang et. al regarding integrating constraints during generation, [2] offers a way to do so without training an HMM, which might integrate nicely with your framework.

References:

[1] Kareem Ahmed, Stefano Teso, Kai-Wei Chang, Guy Van den Broeck, & Antonio Vergari. Semantic Probabilistic Layers for Neuro-Symbolic Learning. NeurIPS 2022.
[2] Kareem Ahmed, Kai-Wei Chang, Guy Van den Broeck. Controllable Generation via Locally Constrained Resampling. In ICLR 2025.

---

> ### Author Response · Authors · 2025-11-21
>
> We thank the Reviewer for their thoughtful and thorough feedback. We are extremely happy that the reviewer found our paper to be well-written, that they appreciated our principled framework that encompasses previous works and that they highlight our speed-up among the strengths of our paper.
>
> > [W1] The paper deals with byte-level LLMs which in my opinion greatly limits its scope as it's hard to draw strong conclusion about its performance on sub-word LLMs that are a lot more commonly used by the community.
>
> We agree that having results for subword LLMs would further strengthen our paper. We plan to address this in the discussion period, please see our proposal in the general response. Specifically, **we plan to apply MTPC to byte-fied SOTA LLMs such as Gemma and Llama to demonstrate the generality of our method**. We also remark that the reason why byte-level LLMs are not widespread is precisely because of their slower decoding, which our work significantly accelerates.
>
> > [W2] The paper details the requirement to train the MTPC which by the authors' description is a very arduous process, and could therefore limit adoptability of the proposed approach.
>
> **We respectfully disagree that training MTPC is an arduous process**. In fact, we have significantly simplified training compared to previous work like Basharin et al. We have found that training with the cross-entropy loss is sufficient, while Basharin et al required mixing the cross-entropy loss with a KL loss as well as an auxiliary loss to help keep the mixture coefficients balanced.
>
> In particular, in preliminary experiments we retrofitted the Llama model Basharin used in their experiments (subword vocabulary size of 50272) and set n=2 and r=2. For this setup computing the KL loss slowed down training by 2x and required more than 2x the memory (Cross-entropy alone required ≈ 30Gb of GPU RAM while including KL required ≈ 70Gb of GPU RAM for a maximum context length of 256) and did not improve our results. Furthermore, many MTP / speculative decoding papers fine-tune the output layer (e.g. Medusa, Hydra). Hence we believe this is not an unreasonable price to pay to obtain a speed up of 1.22x; future work can also build off our setup and insights (e.g. HMM initialisation) to simplify training even further.
>
>
> We address the reviewers questions in the same order as asked:
>
> 1. Yes, MTPC follows the AR LLM distribution. The reason we need to highlight this is that methods like Medusa use a different version of speculative decoding (typical decoding) which is lossy, i.e. does not enforce the guarantee. The “in expectation for sampling” _is not a weakening of that assumption and is well established in the literature_, we use this phrasing to distinguish from greedy decoding where the outputs are guaranteed to be the same.
>
> 2. We always change the MTP output layer of EvaByte and for RQ3 we also adapt the last 1-4 transformer layers, so a part of the model is adapted in any case.
>
> 3. In Section 2 we briefly introduce the needed background / components for our approach. We separate out fully-factorised and CP factorisation because these are methods that have previously been explored in the literature for multi-token prediction. Speculative decoding is not specific to MTP. In fact, the papers that introduced speculative decoding did not use MTP but rather a smaller language model as the draft. As such, we introduce this concept separately.
>
> 4. We thank the reviewer for the pointer. Indeed we use conditional circuits in the same way as the Semantic Probabilistic Layers paper uses conditional circuits. We have included a citation in the updated version of the paper (see **line 252**).
>
> 5. We thank the reviewer for highlighting this discrepancy. Greedy decoding here refers to how the authors of EvaByte evaluated their model on downstream benchmarks, and not on how we evaluated our models on throughput. To address the reviewer’s comment we have: a) moved the details of EvaByte evaluation on benchmarks to the appendix (see **lines 842-871**), to avoid confusion. b) We updated the Training paragraph to include the reasoning for using EvaByte SFT as the verifier by highlighting Medusa’s lossy speculative decoding instead of mentioning greedy decoding, which was confusing. We also now explicitly state that STP has n=1 (see **lines 301-305**). We also plan to release greedy decoding results for throughput evaluation, which will strengthen the paper and further clarify the experimental setup.
>
> 6. We thank the reviewer for the additional pointer, it is indeed relevant! We have included the citation in the conclusion section in the updated paper (see **line 481**).

---

### Author Response · Authors · 2025-11-21
**Updated paper version and additional experiments over the discussion period**

We thank the reviewers for their comments and suggestions. We are excited that their reviews highlight our **elegant and principled MTPC framework that encompasses previous work**, our **rigorous and clear analysis of the trade-off between efficiency and expressiveness**, and our **substantial throughput improvements**. We have taken onboard their feedback and **we have updated the pdf version of the paper on Openreview where required, please see individual responses**.

## Actions
We have also taken onboard their request for additional experiments involving subword-level models \[R1, R2, R3 and R4] and a memory footprint analysis \[R3]. **We will include the following larger changes in the second updated pdf version which we will upload at the end of the discussion period (Dec 3rd):**

## Additional Experiments over Discussion Period

1. **We will retrofit SOTA subword-level models such as Llama 3B and Gemma that have been byte-fied** . While we agree with reviewers that byte-level models are not as mainstream as subword-level models, research on byte-level architectures is very promising \[2,3] and may become mainstream once a key drawback is resolved: generation time which is too slow compared to subword level models. This is why MTP for byte-level models is even more crucial than for subword models: our work has the potential to make byte-level models more competitive. Furthermore, Minixhofer et al. \[1] recently showed that subword-level LLMs (such as the Gemma and Llama family) can be turned into byte-level LLMs with no significant loss of quality. We will therefore retrofit these widely used models using our byte-level LLMs, while at the same time satisfying the requirement that we report results on models of a different size (3B instead of 6.5B).

2. **We will also include greedy speculative decoding results** \[4] for completeness, to highlight that our approach is not limited to sampling and to reduce friction when understanding our experiments and results. From our preliminary experiments we have observed that when doing argmax instead of sampling we obtain a similar boost in throughput.

## Additional Analysis over Discussion Period

1. **We will include a memory footprint analysis** for models during speculative decoding to show what effect $n$, $r$ and the circuit choice has on the GPU memory required. From preliminary results we observe that memory scales sensibly with rank and window size: our speculative decoding code is optimised to share activations and model parameters where possible and all models we trained fit on an NVIDIA 3090 GPU with 24 Gb of RAM.


**We believe that our detailed responses and the additional experiments we will carry out during the discussion period (deadline 3rd Dec) to further strengthen our paper will satisfy our reviewers and encourage them to increase their ratings.**


### References

* [1] Cross-Tokenizer Distillation via Approximate Likelihood Matching <https://arxiv.org/abs/2503.20083>
* [2] Byte Latent Transformer: Patches Scale Better Than Tokens <https://arxiv.org/abs/2412.09871>
* [3] Dynamic Chunking for End-to-End Hierarchical Sequence Modeling <https://arxiv.org/abs/2507.07955>
* [4] Blockwise Parallel Decoding for Deep Autoregressive Models <https://arxiv.org/abs/1811.03115>

---

> ### Author Response · Authors · 2025-12-03
> **Additional Experiments: Llama Model Results and GPU Memory Analysis**
>
> We have carried out the additional experiments and analysis we promised in our initial response to address our reviewers’ questions. Our changes are highlighted in blue in the new version of our paper, see *Appendix J*.
>
> ## Byte-fied Llama 3B results
>
> **Our Llama 3.2 3B results corroborate our findings**. For RQ1, we see a similar increase in acceptance rate and throughput as we increase the rank of our MTP-CP model. For RQ2, we find that BTree is the best performing model even for n=8 in this case, **improving throughput by ×1.23 over MTPC-FF and ×2.07 over STP**. As promised, in addition to speculative sampling, we also report **greedy speculative decoding** results where our MTPC models further **increase throughput by ×1.28 over MTPC-FF and ×3.58** over STP.
>
> From our current understanding, the above trends will also be confirmed when we extend to n=16 and fine-tune additional LoRA layers. We are preparing these additional results for the camera ready version.
>
>
> ## GPU memory consumption
>
> We measured the GPU memory consumption for our models during speculative decoding, using the same settings as our throughput runs (KV cache enabled, BF16, 1,024 generated tokens, on a single NVIDIA H100 80GB GPU). The GPU memory required by the fully factorized head is approximately 13.21-13.27 GB. BTree/HMM with r=32 stays below 13.86 GB (n=8) and 14.57 GB (n=16), i.e., roughly +1.3 GB over the FF baseline. CP scales roughly linearly with rank: at n=8, GPU memory usage increases from 13.35 GB (r=8) to 15.67 GB (r=128); at n=16, it increases from 13.50 GB (r=8) to 18.19 GB (r=128). Retrofitting LoRA on the last 1-4 layers barely changes the footprint: we pay a memory overhead of approximately k transformer layers when using k LoRA layers.

---

### Meta-Review · Area_Chair_gJAR · 2026-01-12

**Summary:**

The paper studies multi-token prediction (MTP) for byte-level LLMs. The main contribution is introducing rich dependencies into the MTP prediction head using the tool of probabilistic circuits (PCs) for better prediction accuracy in speculative decoding (SD). The paper conduct experiments on byte-level LLMs and show that their method is about to improve throughput compared to existing MTP methods.

The reviewers liked the principled approach and analysis that the proposed algorithm is based on. The main concern from the reviewers is regarding the scope of the paper and the comparison to baselines. The paper only evaluates their method on byte-level LLMs and it is unclear how the method can be extended to the mainstream sub-word level LLMs due to its dependence on the vocabulary size. While the authors added experiments on byte-fied LLama and Hulu models, the generalization to subword LLMs is not addressed. Moreover, the paper didn't provide comparison to the important baseline of EAGLE MTP method, which is the SOTA method for subword-level LLMs, and can be used in the byte-level setting as well. This weakens it contribution for byte-level LLMs.

Hence I would recommend the paper for a rejection and I recommend the authors to incorporate reviewers' suggestions to revise the draft for a future submission.

**Reviewer Concerns:**

The authors addressed most of the reviewer concerns except for the limited scope of the proposed method. See above discussion.

**Reviewer Scores:**

I wouldn't expect reviewers' score to change.

---

### Decision · Program_Chairs · 2026-01-26

Reject